# Autophagy and Its Association with Macrophages in Clonal Hematopoiesis Leading to Atherosclerosis

**DOI:** 10.3390/ijms26073252

**Published:** 2025-04-01

**Authors:** Shuanhu Li, Xin Zhou, Qinchun Duan, Shukun Niu, Pengquan Li, Yihan Feng, Ye Zhang, Xuehong Xu, Shou-Ping Gong, Huiling Cao

**Affiliations:** 1Key Laboratory of Pharmacodynamics and Material Basis of Chinese Medicine of Shaanxi Administration of Traditional Chinese Medicine, Engineering Research Center of Brain Health Industry of Chinese Medicine, Pharmacology of Chinese Medicine, Shaanxi University of Chinese Medicine, University Government Committee of Shaanxi Province, Xianyang 712046, China; shuanh-li73@xiyi.edu.cn; 2Xi’an Key Laboratory of Basic and Translation of Cardiovascular Metabolic Disease, Xi’an Key Laboratory of Autoimmune Rheumatic Disease, College of Pharmacy, Xi’an Medical University, Xi’an 710021, China; shuku-niuh-436@xiyi.edu.cn (S.N.); penggqu-li021@xiyi.edu.cn (P.L.); yihan-fen870@xiyi.edu.cn (Y.F.); ye-zhangs773@xiyi.edu.cn (Y.Z.); gongshouping@xiyi.edu.cn (S.-P.G.); 3Laboratory of Cell Biology, Genetics and Developmental Biology, College of Life Sciences, Shaanxi Normal University, Xi’an 710062, China; duanqinchun2020@snnu.edu.cn (Q.D.); or xhx070862@163.com (X.X.)

**Keywords:** atherosclerosis, chaperone-mediated autophagy (CMA), clonal hematopoiesis (CH), programmed cell death, inflammatory, macrophage

## Abstract

Atherosclerosis, a chronic inflammatory disease characterized by lipid accumulation and immune cell infiltration, is linked to plaque formation and cardiovascular events. While traditionally associated with lipid metabolism and endothelial dysfunction, recent research highlights the roles of autophagy and clonal hematopoiesis (CH) in its pathogenesis. Autophagy, a cellular process crucial for degrading damaged components, regulates macrophage homeostasis and inflammation, both of which are pivotal in atherosclerosis. In macrophages, autophagy influences lipid metabolism, cytokine regulation, and oxidative stress, helping to prevent plaque instability. Defective autophagy exacerbates inflammation, impairs cholesterol efflux, and accelerates disease progression. Additionally, autophagic processes in endothelial cells and smooth muscle cells further contribute to atherosclerotic pathology. Recent studies also emphasize the interplay between autophagy and CH, wherein somatic mutations in genes like *TET2*, *JAK2*, and *DNMT3A* drive immune cell expansion and enhance inflammatory responses in atherosclerotic plaques. These mutations modify macrophage function, intensifying the inflammatory environment and accelerating atherosclerosis. Chaperone-mediated autophagy (CMA), a selective form of autophagy, also plays a critical role in regulating macrophage inflammation by degrading pro-inflammatory cytokines and oxidized low-density lipoprotein (ox-LDL). Impaired CMA activity leads to the accumulation of these substrates, activating the NLRP3 inflammasome and worsening inflammation. Preclinical studies suggest that pharmacologically activating CMA may mitigate atherosclerosis progression. In animal models, reduced CMA activity accelerates plaque instability and increases inflammation. This review highlights the importance of autophagic regulation in macrophages, focusing on its role in inflammation, plaque formation, and the contributions of CH. Building upon current advances, we propose a hypothesis in which autophagy, programmed cell death, and clonal hematopoiesis form a critical intrinsic axis that modulates the fundamental functions of macrophages, playing a complex role in the development of atherosclerosis. Understanding these mechanisms offers potential therapeutic strategies targeting autophagy and inflammation to reduce the burden of atherosclerotic cardiovascular disease.

## 1. Introduction

Atherosclerosis is a chronic inflammatory disease of the arterial wall, primarily driven by the accumulation of lipids and immune cells, such as macrophages, which contribute to plaque formation and progression. It remains the leading cause of cardiovascular disease (CVD), resulting in high morbidity and mortality worldwide. While traditionally associated with lipid metabolism disorders, recent advances in understanding atherosclerosis (AS) have highlighted the critical role of cellular processes such as autophagy and inflammation in disease pathogenesis [1,2]. Autophagy, a vital cellular process responsible for maintaining homeostasis, enables the degradation and recycling of damaged proteins and organelles. In macrophages, autophagy plays an essential role in regulating immune responses, lipid metabolism, and the inflammatory processes that are central to atherosclerotic plaque formation [3,4].

Macrophages in atherosclerotic lesions exhibit complex phenotypic changes, including polarization into pro-inflammatory M1 macrophages that exacerbate plaque instability and anti-inflammatory M2 macrophages that promote tissue repair [5,6]. These dynamic macrophage responses are further modulated by cellular pathways like autophagy, which not only maintains macrophage function but also influences the progression of atherosclerosis through its effects on lipid metabolism, cytokine production, and cellular senescence [7,8,9]. Autophagy defects in macrophages have been associated with increased oxidative stress, inflammatory cytokine release, and plaque rupture, thereby accelerating atherosclerosis progression [2,10].

Emerging evidence suggests that autophagy also modulates the fate of vascular smooth muscle cells (VSMCs) and endothelial cells (ECs) within the atherosclerotic microenvironment. For instance, lipophagy, a form of selective autophagy, regulates lipid accumulation in macrophages, facilitating cholesterol efflux and preventing foam cell formation [11,12,13]. However, atherosclerotic conditions impair autophagic activity, which leads to macrophage dysfunction, increased inflammation, and the destabilization of plaques [14,15,16]. This dysfunction is believed to be a critical factor in the progression of atherosclerosis and its complications [17], including myocardial infarction and stroke.

In addition to autophagy, clonal hematopoiesis (CH), a process whereby somatic mutations in hematopoietic stem and progenitor cells lead to the expansion of specific clones, has recently emerged as an important factor contributing to the development and progression of atherosclerosis. Mutations in genes such as ten-eleven translocation-2 (TET2), Janus kinase 2 (JAK2), and DNA methyltransferase 3 alpha (DNMT3A), which are frequently observed in patients with CH, have been shown to promote inflammatory responses and alter macrophage function in the context of atherosclerosis [18,19,20]. These genetic mutations lead to changes in immune cell activity and exacerbate the inflammatory processes underlying plaque formation, highlighting a potential intersection between genetic risk factors and inflammation in atherosclerosis [10].

Given the complex interplay between autophagy, inflammation, and CH in atherosclerosis (Figure 1), targeting these pathways holds promise for developing novel therapeutic strategies aimed at reducing plaque formation, preventing plaque rupture, and improving clinical outcomes in CVD [21,22]. Building upon recent advances in the field, we hypothesize that autophagy, cellular programmed cell death, and clonal hematopoiesis form a critical intrinsic axis, modulating the basic functions of macrophages, which in turn plays a multifaceted and complex role in the development and progression of atherosclerosis. Specifically, this axis is hypothesized to regulate macrophage homeostasis, inflammatory responses, and lipid metabolism, all of which are key drivers in atherosclerotic plaque formation and plaque instability. Moreover, autophagic processes in circulating and CH developed/maturated macrophages are essential not only for cellular maintenance and inflammation resolution but also for preventing exaggerated immune responses and cellular senescence. When this axis is dysregulated, as seen in the context of autophagy defects or somatic mutations associated with clonal hematopoiesis, macrophages are prone to heightened inflammatory states and abnormal lipid accumulation, leading to accelerated atherosclerosis progression. These insights suggest that a more integrated understanding of the molecular underpinnings linking autophagy and clonal hematopoiesis could reveal new therapeutic targets for managing atherosclerotic cardiovascular disease. Recent studies highlight the pharmacological modulation of autophagy as a promising strategy to restore macrophage function and stabilize atherosclerotic plaques. For instance, metformin activates autophagy and suppresses NLRP3 inflammasome activity, reducing plaque progression [23]. Chlorogenic acid enhances macrophage polarization and inflammation resolution by inducing autophagy, while rapamycin improves plaque stability by activating autophagy in macrophages [24,25,26]. Curcumin, a natural compound, also promotes autophagy in macrophages, alleviating atherosclerosis in animal models [27]. These findings, plus those discussed in this review, suggest that targeting autophagy could offer potential therapeutic benefits in preventing or treating atherosclerosis by improving macrophage function and plaque stability.

## 2. Chaperone-Mediated Autophagy (CMA) Initiates in Inflammation and Atherosclerosis

Chaperone-mediated autophagy (CMA) has emerged as a critical process connecting cellular stress responses to inflammation in the context of atherosclerosis. As a specialized form of autophagy, CMA selectively degrades damaged or misfolded proteins containing the KFERQ-like motif (Lys-Phe-Glu-Arg-Gln), which is often present in pro-inflammatory and stress-related proteins [28,29]. In macrophages, CMA plays a pivotal role in modulating the inflammatory environment within atherosclerotic plaques, influencing both plaque stability and disease progression [30]. Research indicates that impaired CMA activity in macrophages can exacerbate inflammatory responses by increasing the secretion of pro-inflammatory cytokines and the activation of inflammasomes, thereby accelerating atherosclerotic development [29]. Furthermore, CMA regulates lipid metabolism, cellular senescence, and oxidative stress, which are processes that are integral to the pathophysiology of atherosclerosis [31]. Expanding on recent findings, in conjunction with others from diverse cellular processes including other forms of autophagy, programmed cell death, and clonal hematopoiesis, CMA establishes an intricate regulatory network that governs macrophage function in atherosclerosis, in which macrophages can be resident from circulating blood and developed/matured CH. This network plays a critical role in macrophage homeostasis, inflammatory regulation, and lipid metabolism, all of which are essential for the development and progression of atherosclerosis. Dysregulation of this axis—whether through impaired CMA, autophagic dysfunction, or mutations related to clonal hematopoiesis—can amplify inflammation and lipid accumulation, accelerating disease progression. These insights highlight the potential for CMA as a therapeutic target for modulating inflammation and plaque stability, offering novel opportunities for therapeutic intervention aimed at controlling inflammation and stabilizing plaques.

### 2.1. Chaperone-Mediated Autophagy as a Specialized Form of Autophagy

Autophagy is a crucial catabolic process in cells, responsible for the degradation and recycling of cellular components such as proteins, lipids, nucleic acids, and organelles. This process not only maintains intracellular homeostasis but also supports cellular renewal by generating new components and energy [28]. Autophagy is broadly categorized into three forms: macroautophagy, microautophagy, and CMA. Unlike macroautophagy and microautophagy, CMA is a highly selective degradation pathway that specifically targets proteins containing the KFERQ motif [32].

In CMA, substrate proteins carrying the KFERQ motif are recognized by heat-shock cognate protein 70 (HSC70), which binds to the substrate and facilitates its translocation to lysosomes via lysosome-associated membrane protein type 2A (LAMP-2A). The protein is then unfolded, and the translocation complex is formed. HSC70 mediates the translocation of the substrate into the lysosome, where lysosomal proteases degrade it. Once degradation is complete, LAMP-2A dissociates from the complex, concluding the CMA process [33,34].

In the context of atherosclerosis, CMA may offer protective effects through several mechanisms. By degrading damaged proteins and organelles, CMA reduces oxidative stress and modulates inflammatory responses. For example, CMA helps prevent plaque progression by degrading oxidized low-density lipoprotein (ox-LDL) in foam cells [35]. Additionally, CMA has been shown to regulate the inflammatory response by inhibiting the secretion of pro-inflammatory cytokines, such as interleukin-1β (IL-1β), and controlling the activation of the NLRP3 inflammasome [29,36].

### 2.2. The Protective Role of Chaperone-Mediated Autophagy in Atherosclerosis

Chaperone-mediated autophagy (CMA) plays a critical protective role in atherosclerosis by modulating cellular homeostasis, particularly within macrophages, which are key players in the disease’s pathogenesis. A 2021 study using ApoE^−/−^ mice and and macrophage-specific conditional LAMP-2A knockout mice revealed that CMA activity is significantly impaired during atherosclerosis progression. The study demonstrated reduced levels of LAMP-2A, a key marker of CMA, which correlated with increased secretion of pro-inflammatory cytokines IL-1β and IL-18. These cytokines, which are dependent on the NLRP3 inflammasome, were found to be elevated due to the impaired CMA-mediated degradation of NLRP3. Notably, the NLRP3 protein contains five KFERQ-like motifs that allow it to interact with HSC70 and LAMP-2A, which are essential components of CMA. This interaction is observed not only in macrophages exposed to lipopolysaccharide (LPS) and ATP but also in human coronary atherosclerotic plaques [29]. Thus, CMA-lysosomal degradation of NLRP3 plays a crucial role in reducing NLRP3 protein levels, which, in turn, helps to limit inflammation and atherosclerosis progression.

While CMA’s protective role has been studied in various diseases [30,37,38], its specific impact on atherosclerosis has only recently come under investigation. This research highlights the potential for pharmacologically activating CMA as a strategy to enhance NLRP3 inflammasome degradation, thus reducing inflammation and slowing atherosclerosis. This approach offers a promising avenue for the development of therapeutic interventions aimed at alleviating atherosclerotic disease [29].

Further corroborating CMA’s protective role in atherosclerosis, a 2022 study demonstrated that CMA inhibition exacerbates cardiovascular disease (CVD) in transgenic mouse models [39,40]. These studies, using KFERQ-PS-Dendra2 mice—expressing a fluorescent reporter for CMA—found that as atherosclerosis progressed, the fluorescence associated with CMA activity in plaques became nearly undetectable. In the systemic CMA-blocked LAMP-2A-deficient (*L2AKO*) mice, researchers showed that CMA deficiency worsens the metabolic and coagulation abnormalities induced by a high-cholesterol diet (Western-type diet, WD), making the mice more autophagy susceptible to atherosclerosis [35,41]. These findings underline that reduced CMA activity in atherosclerotic conditions accelerates disease progression, while promoting CMA could potentially mitigate its severity, presenting a valuable intervention strategy.

Macrophages, known for their pivotal role in atherosclerotic plaque dynamics, express high levels of key CMA effectors, including LAMP-2A and HSC70 [35]. When CMA is impaired, macrophages adopt a more pro-inflammatory phenotype, enhancing their contribution to plaque formation and inflammation. Moreover, CMA inhibition increases the vulnerability of vascular smooth muscle cells (VSMCs) to lipotoxicity, promoting their dedifferentiation and contributing to the autophagic pathological transformation of plaques [42,43]. These observations suggest that CMA is integral to maintaining macrophage homeostasis, modulating inflammatory responses, and protecting against lipid-induced damage in VSMCs. By regulating both macrophage polarization and VSMC function, CMA exerts a protective role in atherosclerosis progression.

Collectively, these studies highlight the importance of CMA in controlling lipid metabolism, inflammation, and cellular responses in the vascular wall. Activating CMA could offer a promising therapeutic strategy to modulate macrophage inflammatory status, prevent foam cell formation, and reduce plaque instability, thus halting the progression of atherosclerosis. This understanding opens the door to potential therapies aimed at enhancing CMA activity to curb cardiovascular disease.

## 3. Deficient Chaperone-Mediated Autophagy Promotes Lipid Accumulation in Macrophages

Macrophages play a pivotal role in the pathogenesis of atherosclerosis, primarily through their ability to internalize lipids and transform into foam cells, which contribute to plaque formation and instability. Lipid accumulation in macrophages is influenced by various processes, including lipid uptake, synthesis, and degradation [44,45]. Autophagy, particularly Chaperone-Mediated Autophagy (CMA), is critical for regulating lipid homeostasis by modulating lipid droplet turnover and degrading lipid-associated proteins. Dysfunction in CMA can lead to impaired lipid metabolism, promoting excessive lipid accumulation and accelerating atherosclerosis progression [46]. This disruption in lipid handling has been linked to altered macrophage function and inflammatory responses, further complicating atherosclerotic disease progression [47,48,49]. In this section, we examine the role of lipids in macrophage function and their contribution to atherosclerotic plaque development, with a particular focus on the impact of CMA dysfunction on lipid metabolism and foam cell formation (Table 1).

### 3.1. Macrophages and Their Fundemental Functions

Macrophages are highly versatile immune cells that perform a wide range of functions, critical to both homeostasis and disease pathology. They are able to adopt distinct phenotypic and functional states depending on the signals present in their local microenvironment. As central players in the immune response, macrophages maintain tissue homeostasis, regulate inflammation, and contribute to tissue repair by engulfing pathogens, cellular debris, and apoptotic cells, as well as secreting a diverse array of cytokines to modulate immune reactions [97].

Morphologically, macrophages are large, irregularly shaped cells with a highly adaptable, amoeboid form. Their cytoplasm is rich in organelles, including lysosomes, mitochondria, and rough endoplasmic reticulum, which are essential for their phagocytic activity and various other cellular processes. These cells are identified by surface markers such as CD14, CD40, CD11b, CD64, F4/80, EMR1, lysozyme M, MAC-1/MAC-3, and CD68, which serve to define their unique role in immune surveillance and response [98,99].

Macrophages exhibit remarkable plasticity, being classified into two primary polarization states: M1 and M2. M1 macrophages are typically induced by pro-inflammatory stimuli such as lipopolysaccharides (LPS) and interferon-gamma (IFN-γ). These cells are characterized by the production of pro-inflammatory cytokines, including tumor necrosis factor-alpha (TNF-α) and interleukin-1 beta (IL-1β), as well as reactive oxygen species (ROS), all of which contribute to inflammation and tissue damage [100]. On the other hand, M2 macrophages are driven by anti-inflammatory signals like interleukin-4 (IL-4) and IL-13, and they are primarily involved in tissue repair, wound healing, and the resolution of inflammation. M2 macrophages release anti-inflammatory cytokines such as IL-10 and facilitate the clearance of apoptotic cells [100].

Furthermore, macrophages exhibit functional diversity based on their anatomical location and activation status. For example, alveolar macrophages in the lungs play a crucial role in clearing inhaled pathogens and particulate matter, thereby contributing to pulmonary defense [101,102]. Kupffer cells in the liver are responsible for detoxifying metabolic byproducts and removing toxins from the bloodstream [103]. Microglia in the central nervous system monitor brain homeostasis, clear dead cells, and respond to injury. Inflammatory macrophages, recruited to sites of infection or injury, are classified into M1 and M2 subtypes based on their functional profiles and cytokine production [104,105].

Vascular macrophages represent a specialized subset of macrophages that reside in or are closely associated with blood vessels, where they play key roles in maintaining vascular integrity and homeostasis. These cells originate from multiple sources, including yolk sac progenitors, bone marrow-derived monocytes, and tissue-resident progenitors [106]. The developmental origins of vascular macrophages are complex, with embryonic yolk sac-derived CX3CR1+ endothelial microparticles (EMPs) and fetal liver monocytes contributing to the establishment of the tissue-resident macrophage population in the arterial wall during early development [107]. Vascular macrophages can be further classified into distinct subtypes, including M1, M2, Mox, M4, and Mhem macrophages, each with unique gene expression profiles and functional roles. For example, Mox macrophages are involved in heme detoxification and mitigating oxidative stress, while M4 macrophages produce chemokines and proteases that recruit additional immune cells and degrade extracellular matrix components [108,109].

In response to vascular injury, inflammation, or pathological conditions such as atherosclerosis or vascular calcification, circulating monocytes infiltrate the vessel wall and differentiate into macrophages. These macrophages play critical roles in vascular pathology, including foam cell formation in atherosclerotic plaques [110], promoting vascular calcification through the secretion of osteogenic factors [111], and facilitating tissue repair through the clearance of apoptotic cells and resolution of inflammation [112]. The plasticity of macrophages allows them to adapt to a variety of microenvironments, making them essential regulators of vascular health and disease.

### 3.2. Lipids Are Essential for Multiple Functions of Macrophages

Lipids play a crucial role in the diverse functions of macrophages, including efferocytosis (the process of clearing apoptotic cells), energy homeostasis, and the regulation of aging [113]. However, the role of lipids in macrophages is context-dependent, with specific lipid species serving different functions based on their cellular localization and the macrophage’s functional state. For example, macrophages can become adipogenic under certain conditions, contributing to diseases such as atherosclerosis [45,47,114]. Recent investigations have also highlighted the importance of lipid accumulation in tumor-associated macrophages (TAMs), lipid-associated macrophages (LAMs) in atherosclerosis, and the selective regulation of lipid metabolism in macrophages [48,115].

In the context of atherosclerosis, macrophage dysfunction is a central event in plaque formation. Macrophages are the predominant immune cell type within atherosclerotic plaques, where they are responsible for lipid uptake, foam cell formation, and the propagation of local inflammation. Macrophages internalize modified low-density lipoproteins (LDLs) and, through the accumulation of lipid droplets (LDs), transform into foam cells [116,117]. The mechanisms underlying lipid accumulation in macrophages involve a combination of altered lipophagy, dysfunction of chaperone-mediated autophagy (CMA), changes in lipid-metabolizing enzymes, and disruptions in lipid uptake and efflux pathways [118].

The autophagy-lysosome-mediated degradation system, including CMA, plays a key role in regulating lipid handling in macrophages. When CMA is disrupted, macrophages fail to properly degrade lipids, resulting in excessive lipid accumulation and foam cell formation, which exacerbates plaque development [119,120]. Recent studies have highlighted the importance of CMA dysfunction in atherosclerosis, with reduced expression of LAMP-2A, a critical component of CMA, linked to lipid accumulation and alterations in lipid-regulating enzymes [49,51].

### 3.3. CMA Plays an Important Role in Lipid Accumulation in Macrophages

Recent studies have established CMA as a crucial regulator of lipid catabolism, primarily by degrading lipid droplet-associated proteins and enzymes involved in lipid metabolism [121,122]. In the absence of CMA, lipid accumulation is observed in various cell types, including hepatocytes in vitro and in mouse liver [121,122]. In macrophages, lipid accumulation is a central event in the progression of atherosclerosis.

Using macrophage-specific *L2AKO* mice and primary peritoneal macrophages from these mice, a 2020 study confirmed that CMA deficiency leads to lipid accumulation in macrophages. This is likely due to the disruption of lipid metabolism-related enzymes. In advanced atherosclerotic lesions, the co-localization of LC3 with lipid droplets (LDs) was reduced, indicating impaired lipid autophagy [51]. Additionally, CMA dysfunction may contribute to lipid accumulation through altered expression of lipid-binding and transport proteins such as scavenger receptor type A (SR-A) and SR-B (CD36), as well as increased expression of long-chain-fatty-acid-CoA ligase 1 (ACSL1), a key enzyme involved in lipid synthesis [51,123]. Concurrently, enzymes responsible for lipid breakdown, such as lysosomal acid lipase (LAL), are upregulated, suggesting that CMA deficiency results in enhanced lipid synthesis and impaired lipid catabolism, further driving lipid accumulation in macrophages [124,125,126]. This disruption of lipophagy in advanced atherosclerotic lesions accelerates lipid buildup and the progression of atherosclerosis [51].

Studies propose that future research should focus on investigating CMA’s role in atherosclerosis using genetic interventions, such as blocking LAMP-2A [29], and the development of pharmacological agents that target LAMP-2A to promote lipid metabolism and alleviate atherosclerosis progression.

### 3.4. PYCARD Modulates CMA-Mediated Lipid Regulation in Macrophages

Recent research has explored the role of PYCARD in regulating chaperone-mediated autophagy (CMA) and its impact on lipid accumulation in macrophages [127,128]. Using pycard knockout (pycard^−/−^) mice and 3T3-L1 cells, a study demonstrated that PYCARD promotes microRNA maturation by inhibiting CMA-mediated degradation of the argonaute *RISC* catalytic subunit 2 (AGO2) [129]. This process plays a key role in regulating neointimal formation following vascular injury, independent of inflammasome activity. The study showed that PYCARD deficiency inhibits miRNA maturation and prevents neointimal formation by promoting CMA-dependent degradation of AGO2, suggesting a novel mechanism through which PYCARD modulates lipid regulation [127,130]. These findings point to the possibility of using PYCARD as a therapeutic target to control lipid accumulation and reduce the progression of atherosclerosis and other vascular diseases.

### 3.5. p62-ATG5-Mediated Autophagy and CMA in Macrophages

Autophagy, a fundamental cellular process for degrading dysfunctional proteins and organelles, plays a critical role in maintaining cellular homeostasis and preventing the accumulation of toxic materials [131,132]. This process is particularly important in macrophages, which are central players in the pathogenesis of atherosclerosis [7,133]. The autophagy pathway includes both bulk autophagy and chaperone-mediated autophagy (CMA), with CMA specifically targeting damaged proteins for lysosomal degradation [134,135]. Disruption of these pathways in macrophages, particularly through deficiencies in key autophagy-related proteins, can exacerbate lipid accumulation and atherosclerotic plaque formation [120,136].

Studies using *p62^−/−^* mice and macrophage-specific ATG5-deficient (*mΦATG5^−/−^*) mice have revealed the importance of p62 in regulating both bulk autophagy and CMA in the context of atherosclerosis. p62 is an autophagic adaptor protein that sequesters polyubiquitinated proteins, directing them to the lysosome for degradation [137]. In macrophages exposed to lipid-rich environments, such as those present in atherosclerosis, p62 accumulates and co-localizes with polyubiquitinated proteins, forming inclusion bodies. These inclusion bodies are typically composed of insoluble protein aggregates and serve a protective role by preventing the cytotoxic effects of protein aggregation [138,139,140].

In macrophages deficient in ATG5, a key protein required for autophagosome formation, the accumulation of p62 and polyubiquitinated proteins is further enhanced. These macrophages exhibit larger cytoplasmic inclusion bodies, which are characterized by an increase in protein aggregation [141]. Notably, in atherosclerotic lesions, the presence of p62-rich inclusion bodies was observed in both mouse aortic plaques and human endarterectomy samples, suggesting that this phenomenon is a hallmark of atherosclerotic disease [138].

The protective role of p62 is critical for the clearance of damaged proteins and the prevention of macrophage dysfunction [142]. In the absence of p62, protein aggregation becomes more pronounced, leading to increased protein insolubility, macrophage apoptosis, and activation of the inflammasome, which in turn promotes the production of pro-inflammatory cytokines such as IL-1β [143]. This inflammatory cascade contributes to the enlargement and increased complexity of atherosclerotic plaques [138].

Given the central role of p62 in managing protein homeostasis through autophagy and its protective effects in macrophages, enhancing p62 function or its mediated clearance pathway presents a promising therapeutic strategy [96]. By boosting the autophagic degradation of toxic protein aggregates and reducing inflammation, such strategies could help mitigate lipid accumulation, plaque instability, and the progression of atherosclerosis.

## 4. Autophagy-Mediated Macrophage Pyroptosis in Atherosclerosis

Atherosclerosis is a complex, multifactorial disease involving various forms of programmed cell death, each playing a distinct role in plaque development and instability. Among these, macrophage pyroptosis has emerged as a key contributor to inflammation and plaque destabilization [144]. Autophagy, a cellular process that helps maintain homeostasis, has been shown to interact with pyroptosis, influencing macrophage survival and inflammatory responses in the context of atherosclerosis [145]. Studies have demonstrated that dysregulated autophagy can exacerbate inflammatory processes in atherosclerotic plaques, while controlled autophagy may protect against excessive pyroptotic cell death [146,147]. This section explores the mechanisms through which macrophage pyroptosis contributes to atherosclerosis progression, particularly in relation to autophagy and other forms of programmed cell death.

### 4.1. Macrophage Pyroptosis

Atherosclerosis is a chronic inflammatory disease characterized by the gradual accumulation of plaque within the walls of medium and large arteries. The development and rupture of plaques in atherosclerosis are associated with damage to vascular cells, including ECs, VSMCs, and macrophages. Autophagy is a subcellular process that influences the pathogenesis of atherosclerosis by regulating the inflammatory response and cell death pathways [148]. Pyroptosis is a newly recognized form of caspase-dependent regulated cell death, characterized by cell swelling, pore formation, and membrane rupture, leading to the massive release of cytoplasmic contents [149]. This mode of cell death is strongly associated with inflammation as it involves the release of pro-inflammatory cytokines, such as IL-1β and IL-18 [150]. In contrast, ferroptosis is a regulated form of non-apoptotic cell death involving iron-dependent lipid peroxidation [151,152]. Programmed cell death encompasses several forms [144,153], including apoptosis, autophagy, pyroptosis, and ferroptosis. In atherosclerosis, these forms of cell death can impact plaque development and stability.

There is increasing evidence that pyroptosis and ferroptosis interact with autophagy and contribute to the development of various diseases, such as cancer, degenerative encephalopathies, and CVD [77,154]. Autophagy can protect macrophages from oxidative stress and inflammatory damage, thereby maintaining cell survival. However, in certain instances, excessive or dysregulated autophagy can lead to cell death, a phenomenon known as “autophagy-dependent cell death” [155]. Although the role of autophagy in atherosclerosis has been extensively studied, the interactions between autophagy, pyroptosis, and ferroptosis, as well as their specific mechanisms in atherosclerosis, require further exploration.

### 4.2. Pathological Studies and Mechanisms of Macrophage Pyroptosis

Pathological studies have identified a significant presence of dead macrophages within vulnerable atherosclerotic plaques, underscoring the association between macrophage death and plaque instability [156]. Macrophage pyroptosis, a pro-inflammatory form of programmed cell death, is induced by activation of the NLRP3 inflammasome and cleavage of caspase-1, both of which are central in triggering pyroptosis [74,157,158]. The activation of caspase-1 leads to the formation of the N-terminal domain of gasdermin D (GSDMD), which then migrates to the plasma membrane to form pores and initiate pyroptosis [159,160,161]. Notably, caspase-1 deficiency has been shown to slow the progression of atherosclerotic plaques in *ApoE^−/−^* mice, underscoring its role in disease advancement [162,163]. ox-LDL has been implicated in aberrant activation of NLRP3, which, in turn, activates GSDMD and exacerbates atherosclerosis in both mouse models and humans [164].

Studies on natural antioxidants have highlighted potential interventions for pyroptosis-related pathways. In 2022, a study using *ApoE^−/−^* mice and human myelomonocytic THP-1 cells found that quercetin effectively activates nuclear factor erythroid 2-related factor 2 (NRF2) by competitively binding to the Arg483 site of kelch like ECH associated protein 1(KEAP1), thereby inhibiting macrophage pyroptosis and reducing oxidative stress levels [165]. These effects were observed at both the cellular and organismal levels, where quercetin administration in high-fat diet-fed *ApoE^−/−^* mice reduced atherosclerosis progression. On the cellular level, quercetin inhibited ox-LDL-induced pyroptosis in THP-1 macrophages by suppressing NLRP3 inflammasome activation and lowering reactive oxygen species (ROS) levels. Further insights were provided by studies using KEAP1 mutants in THP-1 cells, which revealed that quercetin’s anti-pyroptotic effects were specifically associated with the Arg483 residue of KEAP1, not Arg415 [165]. This study suggests that quercetin’s modulation of the KEAP1/NRF2 interaction could be a promising target for reducing pyroptosis in atherosclerosis.

### 4.3. Mechanistic Insights and Therapeutic Targets of Autophagy-Mediated Macrophage Pyroptosis

Recent studies provide compelling evidence that autophagy plays a pivotal role in regulating gasdermin E (GSDME)-mediated pyroptosis, a form of programmed cell death that contributes significantly to inflammation and plaque instability in atherosclerosis [70]. GSDME-mediated pyroptosis has been shown to be induced by a variety of pro-inflammatory stimuli, including oxidized low-density lipoprotein (ox-LDL) in macrophages, which is a key contributor to atherosclerosis [71]. In 2023, research using GSDME knockout ApoE-deficient mice revealed that the deletion of GSDME resulted in reduced atherosclerotic lesion size and lower inflammatory cytokines such as IL-1β, TNF, MCP-1, and IL-6 [70,71]. Single-cell transcriptomic analysis confirmed that macrophages are the primary cells expressing GSDME in atherosclerotic plaques. This study also demonstrated that ox-LDL triggers the expression of GSDME in macrophages, which activates pyroptosis and inflammatory signaling. Notably, the *STAT3* transcription factor was found to regulate GSDME expression, highlighting a transcriptional mechanism through which GSDME-mediated pyroptosis is upregulated in atherosclerotic lesions [71]. These findings suggest that targeting GSDME-mediated pyroptosis could serve as a novel therapeutic strategy for reducing atherosclerotic plaque burden and inflammation.

Moreover, autophagic regulation of macrophage pyroptosis is also observed in the context of hyperhomocysteinemia (HHcy), a common metabolic disorder associated with atherosclerosis. In a 2023 study, caspase-1-deficient (*Casp1^−/−^*) mice and THP-1-derived macrophages demonstrated that homocysteine promotes macrophage pyroptosis through endoplasmic reticulum (ER) stress and calcium dysregulation, both of which are influenced by autophagic processes. This study showed that hyperhomocysteinemia exacerbates atherosclerosis by increasing the size of atherosclerotic plaques and enhancing inflammatory cytokine production, including IL-1β [166,167]. Interestingly, autophagic flux and ER-mitochondrial coupling were found to play critical roles in this process. When autophagy is disrupted, the accumulated ER stress and calcium imbalances contribute to the activation of the NLRP3 inflammasome, which triggers macrophage pyroptosis [166,168]. Thus, targeting autophagy to modulate ER stress and calcium homeostasis may represent a promising approach for preventing HHcy-induced atherosclerosis progression.

Autophagy-mediated macrophage pyroptosis further exacerbates plaque instability and necrotic core formation in advanced atherosclerotic lesions. Dead macrophages release cellular contents, cytokines, and proteases, which promote inflammation and destabilize plaques, increasing the risk of acute cardiovascular events [76,79]. Pyroptosis, specifically through the GSDMD pathway, plays a crucial role in this process. Initially associated with monocytes, pyroptosis, particularly through GSDMD, is now recognized as a key driver of atherosclerotic plaque instability. Ox-LDL, a critical lipid component in atherosclerosis, induces ROS production, which oxidizes GSDMD and activates the NLRP3 inflammasome [169]. Once activated, GSDMD forms pores in the macrophage membrane, initiating pyroptosis and further amplifying inflammation [170]. This suggests that targeting the GSDMD/NLRP3 pathway could be an effective strategy for preventing macrophage-mediated plaque destabilization.

Moreover, autophagy-mediated macrophage pyroptosis is a significant source of cell death in atherosclerosis. Studies show that inhibition of pyroptosis can reduce atherosclerotic progression [165]. Autophagic processes are essential for maintaining macrophage homeostasis and preventing excessive cell death, thereby limiting the release of inflammatory mediators that promote plaque instability. Inhibition of the NLRP3 inflammasome and pyroptosis has been shown to mitigate the harmful effects of oxidative stress in atherosclerosis, further underscoring the therapeutic potential of autophagic modulation in treating cardiovascular disease [166,171]. As autophagy-mediated regulation of macrophage pyroptosis is a critical determinant of atherosclerosis progression, with significant implications for plaque instability and inflammation. Targeting the pathways that modulate autophagic flux and inflammasome activation may offer new therapeutic strategies to mitigate the detrimental effects of macrophage pyroptosis in cardiovascular disease.

### 4.4. Potential and Risk of Pharmacological Manipulation of Autophagic Pathways

As autophagy is an essential cellular degradation process through which damaged organelles, proteins, and other intracellular debris are sequestered in autophagosomes and degraded by lysosomal enzymes, thereby maintaining cellular homeostasis, pharmacological modulation of the autophagy pathway has garnered attention for its potential therapeutic applications across various diseases in recent years. However, this approach also carries inherent risks that require careful consideration.

In the context of atherosclerosis, autophagy plays a crucial role in both the progression and resolution of the disease. Pharmacologically activating autophagy in macrophages, such as through the use of rapamycin, has been shown to promote the polarization of an anti-inflammatory phenotype, reduce vascular inflammation, and improve vascular remodeling [172]. Furthermore, autophagy plays a key role in lipid metabolism, influencing the formation and resolution of atherosclerotic plaques. For example, autophagic activation can reduce plaque progression by facilitating the removal of lipid droplets within macrophages. In patients with non-alcoholic fatty liver disease (NAFLD), autophagy activation has been demonstrated to alleviate hepatic lipid accumulation. Resveratrol, a small molecule drug, can activate the AMPK pathway and inhibit mTORC1 activity, thereby promoting autophagic flux and regulating autophagy-related gene expression via the PI3K/AKT signaling pathway [172].

Beyond lipid metabolism, autophagy is integral to cellular homeostasis, particularly in neurons. For instance, autophagic activation reduces the accumulation of α-synuclein in the brains of Parkinson’s disease models [173,174]. Additionally, the autophagy-inducing drug trehalose has demonstrated protective effects in animal models of neurodegenerative diseases, highlighting its potential for clinical application [175].

Autophagy also has a dual role in cancer therapy. On one hand, autophagy can promote cancer cell survival and resistance to chemotherapy. On the other hand, autophagy modulation, such as through rapamycin, may enhance the efficacy of cancer therapies. However, excessive autophagy activation can lead to cellular dysfunction, including disrupted energy metabolism and the induction of apoptotic pathways [176]. Dysregulated autophagy may also contribute to lipotoxicity, thereby increasing oxidative stress within cells. In certain contexts, excessive autophagy can exacerbate disease progression, such as in tumors where autophagy supports growth and drug resistance.

While the pharmacological manipulation of autophagy pathways holds considerable therapeutic promise, it is not without its risks. Overactivation of autophagy can impair cellular function, leading to metabolic disturbances and heightened oxidative stress. Conversely, inhibition of autophagy may result in an exacerbated inflammatory response, potentially aggravating disease progression. For instance, chronic administration of rapamycin, although effective in inducing autophagy, has been associated with immunosuppression and metabolic disorders, underscoring the need for caution in its long-term use [177,178].

In summary, while pharmacological regulation of the autophagy pathway holds significant promise for the treatment of various diseases (Table 2), it is essential to evaluate the risks and fine-tune therapeutic strategies to ensure efficacy and minimize adverse effects. Future research should focus on elucidating the precise mechanisms governing autophagy and developing more selective and safer pharmacological agents that can harness the full therapeutic potential of autophagy without undesirable side effects. 

## 5. Macrophage Autophagy and Clonal Hematopoiesis in Atherosclerosis

In adult mammals, circulating macrophages, derived from the monocyte-macrophage lineage, are generated from four primary sources: (1) the yolk sac at embryonic day 7.0 (E7.0), (2) the embryonic liver at E9.5, (3) the dorsal aorta, and (4) the bone marrow. These sources give rise to pre-macrophages/macrophages, monocytes, perivascular stromal cells (PDGFRA^+^), and hematopoietic stem cells (HSCs) through distinct waves of hematopoiesis [191,192]. Upon migration, circulating macrophages can localize to various tissues, including the heart, where atrioventricular (AV) node-resident macrophages directly influence cardiac function, as characterized by electrocardiogram (EKG) abnormalities [193,194,195]. Genetic defects in macrophage function can lead to severe cardiac dysfunction, particularly by disrupting the integrity of the AV node and its associated structures, such as the plane of insulation (POI) adjacent to the central fibrous body (CFB) [194,195]. It is plausible that circulating macrophages with impaired autophagic function localize to the aortic wall, where they play a critical role in the pathogenesis of atherosclerosis. 

Clonal hematopoiesis (CH), driven by somatic mutations in hematopoietic stem cells (HSCs), is increasingly recognized as a significant risk factor for atherosclerosis. Mutations in genes such as *TET2*, *JAK2*, and *DNMT3A* promote the expansion of mutant clones in the blood, leading to inflammation, macrophage dysfunction, and plaque instability [196,197]. These mutations are particularly associated with clonal hematopoiesis of indeterminate potential (CHIP), a condition that accelerates atherosclerosis through the activation of inflammasomes and increased oxidative stress within macrophages [198]. While some studies suggest that CHIP mutations exacerbate atherosclerosis, others argue that CHIP may be a consequence of the chronic inflammatory state associated with atherosclerosis [199,200]. This section explores the genetic and inflammatory mechanisms through which CHIP mutations influence macrophage behavior and contribute to atherosclerotic plaque development.

### 5.1. Clonal Hematopoiesis as a Risk Factor for Atherosclerosis

Clonal hematopoiesis of indeterminate potential (CHIP) is a newly recognized, age-related risk factor for atherosclerosis, driven by somatic mutations in specific leukemia-related driver genes along with other age-related diseases including cardiac diseases [201]. These mutations lead to the expansion of mutant cell clones in peripheral blood. CH occurs when HSCs acquire genetic mutations and differentiate through polylineage hematopoiesis to produce terminally differentiated blood cells carrying clonal markers [202,203]. Common mutations associated with CHIP typically involve one of four genes—*TET2*, additional sex combs like-1 (*ASXL1*), *DNMT3A* or *JAK2*—which promote the clonal expansion of hematopoietic cells [89,187]. Although CH itself is not a malignancy, its presence significantly increases the risk of hematologic cancers. However, this alone does not explain the elevated mortality seen in individuals with CHIP [204]. The relationship between CHIP and atherosclerotic CVD has been explored in various studies. Some research suggests that certain CHIP mutations may directly contribute to atherosclerosis by exacerbating the inflammatory response, while other studies propose that CHIP may be a consequence of atherosclerosis, with the disease accelerating the expansion of mutant clones [18,96,196,205]. Furthermore, mutations commonly observed in CHIP, such as *DNMT3A*, *TET2*, *ASXL1* and *JAK2* [19,206] plus described above, have been linked to arterial disease, such as tubular artery disease.

### 5.2. Mechanisms Linking Clonal Hematopoiesis to Atherosclerosis

Recent studies have elucidated key mechanisms through which clonal hematopoiesis (CH) mutations contribute to atherosclerosis. A study by Fidler et al. demonstrated that mutations in *JAK2*, specifically the *JAK2V617F* mutation, promoted macrophage proliferation and necrotic core formation in atherosclerotic plaques in mutant mice. This process was associated with the activation of the AIM2 inflammasome, which led to an increase in IL-1β production and enhanced oxidative stress [187]. The study also highlighted the critical role of macrophage metabolic reprogramming in CH, with *JAK2* mutations driving the accumulation of reactive oxygen species (ROS) and oxidized DNA [187]. Furthermore, the administration of the IL-1 receptor antagonist anakinra was shown to reduce plaque instability by normalizing macrophage proliferation, reducing necrotic core size, and stabilizing the fibrous cap [207,208,209]. These findings underscore the important connection between JAK2 mutations, inflammasome activation, oxidative damage, and atherosclerosis, suggesting potential therapeutic avenues for managing CHIP-related cardiovascular risk.

In another pivotal study, Bick et al. employed whole-genome sequencing in a cohort of nearly 100,000 individuals to investigate the genetic mechanisms of CHIP. The researchers identified several CHIP-associated mutations in genes such as *DNMT3A*, *TET2*, *ASXL1*, *JAK2*, and *PPM1D*, which were found to be significantly associated with lipid metabolism, immune function, and inflammatory responses [202]. Notably, a variant of *TET2* was identified as being more prevalent in individuals of African ancestry, suggesting that genetic diversity may influence the impact of CHIP on atherosclerosis risk [210,211]. These findings provide important insights into the genetic basis of CHIP and open potential avenues for future diagnostics and therapeutic interventions for CHIP-related cardiovascular disease.

Additionally, research by Fuster et al. investigated the role of *TET2* mutations in both CHIP and atherosclerosis. The loss of *TET2* in hematopoietic stem cells (HSCs) was shown to disrupt DNA methylation patterns, thereby influencing hematopoietic cell clonality [200] and promoting inflammatory processes that accelerate atherosclerosis. *TET2* deficiency led to altered immune responses and increased plaque formation in mouse models, emphasizing the significant role of epigenetic changes in CHIP and its potential contribution to the pathogenesis of atherosclerosis [212,213]. These findings provide critical insights into how genetic mutations in hematopoietic cells influence atherosclerosis development, reinforcing the importance of clonal hematopoiesis as a significant risk factor for cardiovascular disease.

### 5.3. Confirmed Function of Circulating Macrophages in the Cardiovascular System

The heart is composed of a diverse array of cell types, predominantly cardiomyocytes, endothelial cells, fibroblasts, pericytes, smooth muscle cells, and various leukocytes, including macrophages. Traditionally, the immune system has not been closely associated with the heart, but recent studies have shown that immune cells, particularly macrophages, play a vital role in maintaining heart health [191,193,194,214,215].

Using gene network analysis and single-cell regulatory network inference (SCENIC), Sano et al. provided mechanistic evidence that mosaic loss of the Y-chromosome (*mLOY*) in leukocytes can impact cardiac function [191]. Hulsmans et al. further demonstrated that hematopoietic macrophages can localize to the heart, particularly in the atrioventricular (AV) node, where they bind to conductive cardiomyocytes. These resident macrophages are essential for regulating the heart’s electrophysiological activity through the gap-junction protein connexin 43 (Cx43) [195].

These findings confirm that hematopoietic macrophages contribute to arrhythmias commonly observed in patients with heart failure. Specifically, macrophages localized to the AV node disrupt electroconductive gap-junctions, leading to arrhythmias and cardiac fibrosis, as demonstrated in their mouse models [191,195] (Figure 2). In the context of ischemia/reperfusion injury, bone marrow-derived monocytes increase in response to myocardial stress, localizing to macrophage niches within the myocardium, where they aid in tissue repair. This suggests that cardiac resident macrophages could also be pivotal in the development and progression of atherosclerosis.

### 5.4. TET2, JAK2, and DNMT3A-Associated Autophagy Could Lead Clonal Hematopoiesis to Atherosclerosis

The relationship between clonal hematopoiesis (CH) and atherosclerosis is increasingly recognized as a critical factor in cardiovascular disease pathogenesis [216,217]. Mutations in key genes such as *TET2*, *JAK2*, and *DNMT3A* not only promote clonal expansion of hematopoietic stem cells (HSCs) but also alter the function of macrophages, a major cell type in atherosclerotic lesions [200,218]. These mutations impact autophagic pathways, which are essential for macrophage homeostasis, inflammation resolution, and lipid metabolism [90,219]. The role of autophagy in macrophages is particularly pertinent, as its dysfunction contributes to the inflammatory milieu within atherosclerotic plaques [127,219]. Among the various genetic alterations associated with CH, *JAK2* mutations have emerged as a key modulator of both autophagy and inflammation, influencing macrophage behavior and plaque development [218]. This section explores the mechanisms by which *JAK2* mutations disrupt autophagic regulation in macrophages and how this dysregulation accelerates atherosclerosis, setting the stage for understanding potential therapeutic interventions targeting *JAK2* and autophagic pathways in atherosclerosis [71,166].

#### 5.4.1. JAK2 and Autophagic Regulation in Atherosclerosis

Clonal hematopoiesis (CH), driven by mutations in genes such as *JAK2*, has been implicated in the pathogenesis of atherosclerosis, particularly by modulating macrophage function and autophagy. JAK2, a central regulator of immune and inflammatory responses, contributes to autophagic regulation in macrophages. Interleukin-6 (IL-6), a key pro-inflammatory cytokine elevated in atherosclerosis, activates JAK2 signaling and induces the phosphorylation of Beclin 1 (BECN1), a critical protein involved in the initiation of autophagy [218]. This phosphorylation enhances BECN1’s role in autophagy, helping macrophages survive under stress conditions and contributing to inflammation within atherosclerotic plaques. JAK2 activation in macrophages thus not only promotes autophagy but also influences the inflammatory environment, exacerbating macrophage dysfunction and disease progression [218]. This suggests that *JAK2* could be a potential therapeutic target to modulate macrophage autophagy and inflammation in atherosclerosis.

#### 5.4.2. TET2 Mutations and Impaired Autophagy in Atherosclerosis

TET2, a DNA demethylase, plays a critical role in the regulation of macrophage autophagy. Mutations in *TET2*, commonly found in CH, disrupt autophagic flux, especially in the hypoxic microenvironment of atherosclerotic lesions. Hypoxia-induced alterations in TET2 activity lead to changes in DNA methylation patterns, affecting the expression of genes crucial for autophagy. Loss of *TET2* activity results in enhanced macrophage inflammation and reprogramming, which accelerates atherosclerotic lesion formation [90,218,220]. These findings highlight the importance of TET2 in maintaining macrophage function and autophagic homeostasis within the atherosclerotic context, suggesting that *TET2* mutations may contribute to atherosclerosis by disrupting macrophage autophagic processes.

#### 5.4.3. DNMT3A and Epigenetic Regulation of Autophagy in Atherosclerosis

Mutations in *DNMT3A*, a DNA methyltransferase, further complicate the epigenetic regulation of autophagy in macrophages. González-Rodríguez et al. demonstrated that DNMT3A-mediated DNA methylation at the *MAP1LC3* locus—critical for initiating autophagy—leads to reduced autophagic flux [219]. In *DNMT3A*-mutated macrophages, this impairment exacerbates lipid retention and inflammatory responses, accelerating the progression of atherosclerosis. These findings underscore the role of DNMT3A in regulating autophagic processes that modulate macrophage function and inflammation in atherosclerotic disease. Therefore, *DNMT3A* mutations, similar to *TET2* and *JAK2*, link epigenetic regulation of autophagy to macrophage-driven atherosclerosis [218,219].

In conclusion, mutations in *TET2*, *JAK2*, and *DNMT3A* not only drive clonal hematopoiesis but also modulate autophagic pathways that exacerbate the inflammatory microenvironment within atherosclerotic plaques. These genetic alterations contribute to macrophage dysfunction, lipid retention, and plaque instability, highlighting CH as a significant risk factor for cardiovascular disease. The dual role of CH in both the hematopoietic and vascular systems underscores its potential as a therapeutic target in atherosclerosis. Targeting the autophagic regulation modulated by *TET2*, *JAK2*, and *DNMT3A* mutations may offer new avenues for managing atherosclerotic cardiovascular disease.

### 5.5. The Role of the Monocyte-Macrophage System in Cardiovascular Health

In the adult mammalian heart, there are three distinct populations of macrophages, each originating from different waves of hematopoiesis. The first wave originates from embryonic yolk sac macrophages around embryonic day 7.0. The second wave arises from fetal liver monocyte progenitors at embryonic day 9.5. The third wave consists of the traditionally recognized hematopoietic monocyte-macrophage system [221,222,223,224,225] (Figure 2).

Recent studies have highlighted the role of macrophages derived from embryonic progenitors in both myocardial and vascular repair. These macrophages, localized in the myocardium and within the walls of large and medium-sized arterioles, are involved in cardiac function recovery under pathological conditions [226,227,228]. Autophagic dysfunction in these macrophages can significantly affect their role in cardiovascular diseases, including atherosclerosis.

A recent study utilizing long-term in vivo tracing of yolk sac-derived hematopoietic stem cells (HSCs) demonstrated that these cells migrate to the aorta-gonad-mesonephros (AGM) region, umbilical vessels, and other extraembryonic tissues during development. Notably, these progenitors retain their hematopoietic potential even in adult mice [229]. Samokhvalov et al. showed that yolk sac-derived cells expressing *Runx1* at embryonic day 7.5 give rise to fetal lymphoid progenitors and adult HSCs [229].

In a similar vein, Yokomizo et al. used genetic tracing to demonstrate that fetal yolk sac-derived HSCs, expressing the liver-specific transcription factor hepatic leukemia factor (*HLF*), play a critical role in the post-gestational development of intra-arterial hematopoietic clusters, rather than contributing significantly to fetal hematopoiesis [230]. These findings have been validated in zebrafish models [228]. Further evidence of the importance of embryonic HSCs in adult hematopoiesis comes from studies by Wattrus et al., which showed that embryonic macrophages are essential for establishing proper adult hematopoiesis through quality control mechanisms [231].

In aging humans, *mLOY* not only affects cardiac function but also has been linked to carcinogenesis, as the progressive accumulation of *LOY* in cells correlates with aging [232,233]. Thompson et al. proposed that macrophage *LOY* serves as a biomarker of general genetic instability, consistent with *mLOY* across various cell types throughout the body [195,233]. These mutations, particularly in leukocytes including macrophages, may have functional consequences, contributing to age-related diseases such as cancer and cardiovascular disorders. Moreover, mutations in genes involved in autophagy pathways may strongly influence the development of atherosclerosis [201,234,235,236], highlighting the broader implications of autophagic dysregulation in cardiovascular diseases.

## 6. Conclusions and Perspective

Atherosclerosis is a multifaceted, chronic inflammatory disease in which various proteolytic pathways, such as macroautophagy, the ubiquitin–proteasome system (UPS), and chaperone-mediated autophagy (CMA), play pivotal roles. Among these, macroautophagy has been well-characterized in atherosclerosis, while the contribution of CMA remains less understood. CMA is a selective protein degradation process that targets over 45% of cytoplasmic proteins containing KFERQ-like motifs, and it is crucial for maintaining cellular proteostasis and lysosomal function. Dysregulation of CMA can influence macroautophagy and exacerbate atherosclerosis progression. Notably, CMA facilitates the degradation of pro-atherogenic proteins, such as NLRP3 inflammasomes, making it a promising target for therapeutic intervention in atherosclerosis. Despite this potential, further research is required to fully understand how CMA affects NLRP3 inflammasome activity and its broader role in atherosclerotic pathogenesis.

CMA dysfunction, particularly in macrophages and their associated clonal hematopoiesis, results in lipid accumulation, oxidative stress, and inflammatory cytokine release—key contributors to atherosclerosis. While the link between CMA deficiency and disrupted lipid metabolism in either circulating and CH developed/maturated macrophages is established, the precise mechanisms by which CMA regulates lipid accumulation and atherogenesis remain unclear. CMA also plays a vital role in lipid homeostasis by modulating lipid-regulating enzymes, thereby preventing foam cell formation and plaque instability. Emerging evidence suggests that both macroautophagy and CMA contribute to macrophage function and plaque stability, pointing to the potential for therapeutic strategies that target both pathways to mitigate atherosclerosis progression. These findings underscore the necessity of further research into the intricate regulation of lipid metabolism by CMA.

Another critical aspect of atherosclerosis is macrophage pyroptosis, a pro-inflammatory form of cell death that destabilizes plaques. Pyroptosis, mediated by proteins such as GSDMD, exacerbates atherosclerosis by impairing cholesterol transport, with GSDMD knockout models showing reduced lesion areas [237,238]. Additionally, GSDME, another regulator of inflammatory cell death, can switch apoptosis to pyroptosis depending on caspase-3 activity. GSDME’s role in regulating cell death pathways presents a promising therapeutic target for atherosclerosis. Building on recent advances, we hypothesize that CMA, in conjunction with other forms of autophagy, programmed cell death, and clonal hematopoiesis, forms a crucial regulatory axis that governs all-source macrophage function in atherosclerosis. This network plays a pivotal role in regulating macrophage homeostasis, inflammation, and lipid metabolism, all of which are central to the development and progression of atherosclerosis. Disruption of this axis—whether through CMA dysfunction or clonal hematopoiesis—can lead to excessive inflammation, impaired lipid handling, and plaque destabilization. This hypothesis suggests that targeting the key components of this axis may offer a powerful approach for mitigating atherosclerosis and preventing its complications. Understanding the intricate relationships between autophagy, cell death, and clonal hematopoiesis will be essential in designing novel therapeutic interventions for atherosclerotic cardiovascular disease.

In conclusion, the intricate interplay between CMA, lipid metabolism, and pyroptosis represents an untapped opportunity for innovative therapeutic interventions. Targeting these pathways, either individually or synergistically, offers a powerful strategy to slow or even reverse the progression of atherosclerosis, with the potential to significantly impact the management of atherosclerotic cardiovascular disease.

## Figures and Tables

**Figure 1 ijms-26-03252-f001:**
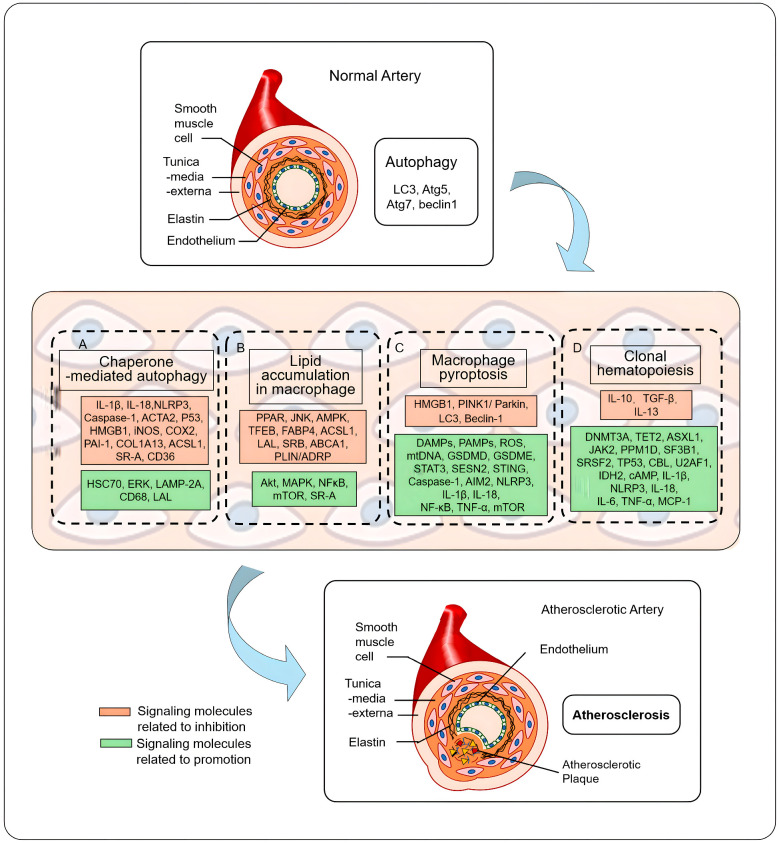
Signaling molecular pathways are altered from circumstance in a normal physiological artery (**top**) to that in an atherosclerotic artery (**bottom**) through chaperone-mediated autophagy, macrophage pyroptosis, and clonal hematopoiesis (**middle**) to atherosclerosis (AS). (**A**) Chaperone-mediated autophagy (CMA) is a selective autophagic process. (**B**) Dysfunction of autophagy affects macrophage lipid accumulation. (**C**) Autophagy mediates pyroptosis in macrophages. (**D**) Mutations in clonal hematopoiesis (CH) are the AS risk factors (figure not to scale).

**Figure 2 ijms-26-03252-f002:**
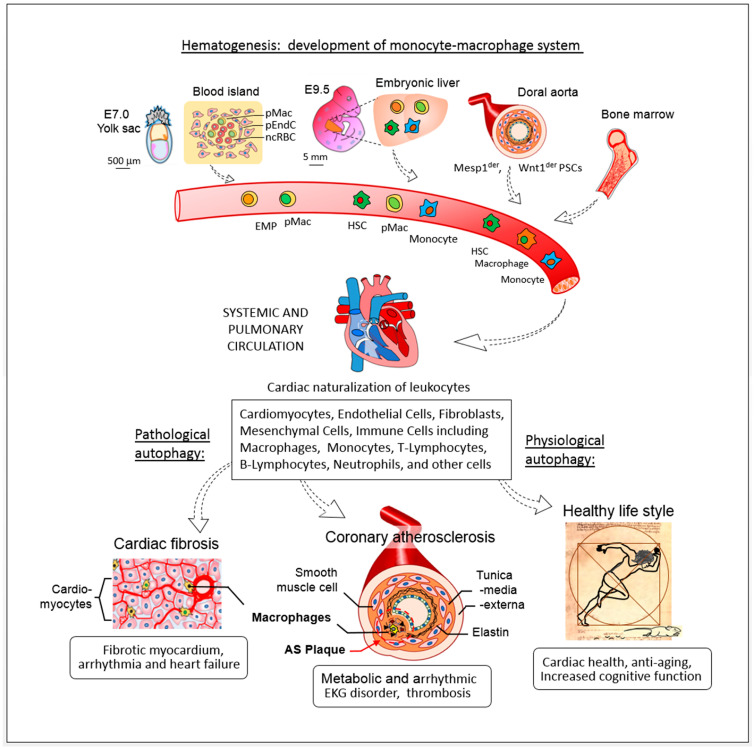
Cellular and developmental mechanisms of macrophages in clonal hematopoiesis associated with atherosclerosis. This figure illustrates the various stem cell lineages involved in the development of macrophages during clonal hematopoiesis (CH) and their contributions to cardiovascular pathophysiology. Macrophages originating from these lineages play a significant role in cardiac fibrosis, heart failure, and overall cardiovascular function. These macrophages arise from hematopoietic stem cells (HSCs), which begin their development from yolk sac cells during the early embryonic stage [139]. Hematopoiesis continues through the embryonic liver and dorsal aorta, ultimately transitioning to bone marrow for sustained blood cell production. Pathological autophagy in these macrophages contributes to cardiovascular diseases, including cardiac fibrosis and coronary ischemia. In contrast, physiological autophagy, promoted by a healthy lifestyle, supports cardiac health and enhances cognitive function, potentially mitigating the effects of aging (figure not to scale).

**Table 1 ijms-26-03252-t001:** Autophagy involved atherosclerosis through inhibition and promotion cellular pathways.

Autophagy Related Cellular Processes	Involved Autophagy Genes for Functional Atherosclerosis			
	Signal Molecules Related to Promotion		Signal Molecules Related to Inhibition	
	Genes Involved	Ref.	Genes Involved	Ref.
Chaperone-Mediated Autophagy	CD68, ERK, HSC70, LAMP-2A, LAL	[33,34,35,50]	ACSL1, ACTA2, COL1A1-3, COX2, CD36, caspase-1, HMGB1, iNOS, IL-1β, IL-18, NLRP3, P53, PAI-1, SR-A	[29,51,52,53,54,55,56,57,58]
Lipid Accumulation in Macrophage	Akt, MAPK, mTOR, NF-κB, SR-A	[56,59]	AMPK, ACSL1, ABCA1, FABP4, JNK, LAL, PPAR, PLIN/ADRP, SR-B, TFEB	[47,60,61,62,63,64,65,66,67]
Macrophage Pyroptosis	AIM2, caspase-1, DAMPs, GSDMD, GSDME, IL-1β, IL-18, mTOR, NLRP3, NF-κB, PAMPs, ROS, STAT3, SESN2, STING, TNF-α	[68,69,70,71,72,73,74,75,76,77,78,79]	beclin-1, HMGB1, LC3, PINK1/Parkin	[80,81,82,83]
Clonal Hematopoiesis	ASXL1, CBL, DNMT3A, cAMP, IDH2, IL-1β, IL-18, IL-6, JAK2, MCP-1, NLRP3, PPM1D, SF3B1, SRSF2, TET2, TP53, TNF-α, U2AF1	[84,85,86,87,88,89,90,91,92,93,94,95]	IL-10, IL-13, TGF-β	[96]

Abbreviation: ABCA1: ATP-binding cassette transporter A1; ACSL1: acyl-CoA synthetase long chain family member 1; ACTA2: Actin Alpha 2; AIM2: Absent in Melanoma 2; Akt: Protein Kinase B; AMPK: Adenosine 5’-monophosphate(AMP)-activated protein kinase; ASXL1: Additional Sex Combs Like Transcriptional Regulator 1; beclin-1: B-cell lymphoma 2-interacting myosin-like coiled-coil protein 1; cAMP: cyclic adenosine monophosphate; caspase-1: Cysteine Aspartate-Specific Protease 1; CBL: Casitas B-lineage Lymphoma; CD36: CD36 molecule (thrombospondin receptor); CD68: Cluster of Differentiation 68; COL1A1-3: Collagen Type I Alpha 1 Chain, Collagen Type I Alpha 2 Chain, Collagen Type I Alpha 3 Chain; COX2: Cyclooxygenase-2; DAMPs: Damage-associated molecular patterns; DNMT3A: DNA methyltransferase 3 alpha; ERK: Extracellular signal-regulated kinase; FABP4: fatty acid binding protein 4; GSDMD: gasdermin D; GSDME: gasdermin E; HMGB1: High mobility group box 1 protein; HSC70: Heat shock cognate protein 70; IDH2: isocitrate dehydrogenase (NADP+)2; IL-10: interleukin 10; IL-13: interleukin 13; IL-18: interleukin-18; IL-1β: Interleukin-1β; IL-6:interleukin 6; iNOS: Inducible Nitric Oxide Synthase; JAK2: Janus kinase 2; JNK: c-Jun N-terminal Kinase; LAL: Limulus Amebocyte Lysate; LAMP-2A: Lysosomal-associated membrane protein 2A; LC3: Microtubule-Associated Protein 1 Light Chain 3; MAPK: Mitogen-Activated Protein Kinase; MCP-1: Monocyte chemoattractant protein-1; mTOR: Mammalian Target of Rapamycin; NF-kB: Nuclear factor kappa-light-chain-enhancer of activated B cells; NLRP3: NOD-like receptor pyrin domain containing 3; P53: Tumor protein p53; PAI-1: Plasminogen Activator Inhibitor-1; PAMPs: Pathogen-Associated Molecular Patterns; Parkin: Parkin RBR E3 ubiquitin-protein ligase; PINK1: PTEN induced putative kinase 1; PLIN/ADRP: Perilipin/Adipose differentiation-related protein; PPAR: Peroxisome Proliferators-Activated Receptors; PPM1D: Protein Phosphatase, Mg2+/Mn2+Dependent 1D; ROS: Reactive Oxygen Species; SESN2: Sestrin2; SF3B1: Spliceosome factor 3b subunit 1; SR-A: Scavenger receptor class A; SR-B: Scavenger Receptor Class B; SRSF2: Serine/Arginine-Rich Splicing Factor 2; STAT3: Signal Transducer and Activator of Transcription 3; STING: Stimulator of interferon genes; TET2: Ten-eleven translocation 2; TFEB: Trancription Factor EB; TGF-β: Transforming Growth Factor Beta; TNF-α: Tumor Necrosis Factor-alpha; U2AF1: U2 Small Nuclear RNA Auxiliary Factor 1.

**Table 2 ijms-26-03252-t002:** Current Treatments and Preclinical/Clinical Data for Atherosclerosis.

Pathway		Drug Targeting Molecule	Agent Product	Biological Process/Mechanism of Action	Preclinical/Clinical Data	Class/Phase	References
Autophagy associated drugs/molecules against atherosclerosis	Lipid Accumulation in Macrophage	SGLT2	Enpagliflozin, Dagliliflozin	Activation of autophagy through the AMPK signaling pathway → clears intracellular lipid accumulation and damaged organelles, reducing inflammatory responses	Improved glycemic control in T2DM and reduces the risk of cardiovascular adverse cardiovascular events	Class II	[76,179]
		Adiponectin receptor agonists	AdipoRon	Activate AMPK → enhance macrophage autophagy →promote cholesterol efflux	Cholesterol efflux increased by 40%, autophagy marker LC3-II/LC3-I ratio increased by 3 times	N/A	[62]
		mTOR	Rapamycin	Inhibit mTOR→activation of autophagy→promote the burial effect	Rapamycin significantly reduced the atherosclerotic lesions in the aorta in a high-fat diet-induced atherosclerosis mouse model	Phase II	[69,180]
		SGLT2	Empagliflozin	Activation AMPK→inhibition of the mTORC1 activity→deregulate the inhibitory effect of mTOR on autophagy	Reduced mortality in patients with type 2 diabetes mellitus complicated with atherosclerotic cardiovascular disease	Phase III	[33,181]
		TBK1/IKKε	BX795	The TBK1/IKKε signaling pathway activates autophagy → clears damaged organelles and lipid accumulation within cells → reduces the production of inflammatory factors	TNF- α and IL-6 secretion were reduced by 70%, and autophagy-related genes (Atg 5) were upregulated	N/A	[182]
	Macrophage Pyroptosis	Colchicine	Colchicine, low dose	Inhibition of NLRP3 inflammasome activation; it may interfere with autophagosome-lysosome fusion	Reduce the risk of cardiovascular events and improve coronary plaque stability	Class II	[74,183]
		NLRP3	Tranilast	Inhibition of the NLRP3 protein →reduce in IL-1 β release; Increase the LC3-II/LC3-I ratio and reduce p62 accumulation	Significantly reduced the atherosclerotic plaque formation in the aorta	Phase II	[78]
		Caspase-1	VX-765	Blocking caspase-1 downstream of the inflammasome → inhibition of the GSDMD-mediated pyroptosis	The plaque pyroptotic cells were decreased by 55%, and the plaque stability was enhanced	N/A	[67]
		Gasdermin	Disulfiram	Blocking of the gasdermin D pore tract → inhibition of pyroptosis	Inhibition of pyroptosis reduced the plaque necrotic core	Phase I	[184]
		GSDME	Gasdermin E inhibitor	Blocking of the GSDME shearing →inhibit pyrodead pore formation	Plaque area decreased by 30%, fibrous cap thickness increased by 50%, and 60% decreased macrophage pyroptosis markers (caspase-3 activity)	N/A	[80,185]
	Clonal Hematopoiesis	JAK2	Fedratinib	Inhibition of the expansion of the JAK 2 mutant clones → reduction of the proinflammatory macrophages	Reduce IL-6, and TNF- α secretion	N/A	[185,186,187]
		TET2	N/A	Reduce the methylation level of the Beclin1 promoter region → promote the expression of Beclin1	Increase the expression of autophagy marker LC3-II, while reducing the accumulation of autophagy disorder marker p62	N/A	[81]
Drugs or molecules of other pathways against atherosclerosis		Statins	Atorvastatin, Rosuvastatin	Inhibits HMG-CoA reductase → lowers LDL-C	LDL-C decreased to <100 mg/dL	Class I	[188]
		PCSK9	Alirocumab, Elosulfase	Blocking of the PCSK 9→increasing the LDL receptor expression →reduce LDL-C	Lower plasma LDL-C levels by 50–60%	Class I	[189]
		IL-1β	Canakinumab	Neutralization of IL-1 β→inhibition of the NLRP3 inflammasome	Significant reduction in the hs-CRP levels	Class II	[29]
		Antiplatelet drugs	Aspirin, Clopidogrel	Inhibition of platelet aggregation	Could significantly reduce the risk of stent thrombosis and cardiovascular events	Class I	[87,183]
		GPX4	RSL3	Triggering of macrophage iron death by induction of lipid peroxidation →reduce intraplaque necrotic core	The plaque necrotic core was reduced by 40%	N/A	[67,189]
		NLRP3/TRPV2	Tranilast	Inhibition of the NLRP3 inflammasome blocks the TRPV 2 channels → synergy anti-inflammatory	Spot collagen content increased by 25% and macrophage apoptosis decreased by 40%	N/A	[78]
		CCR2	RS504393	Blocking CCR2 → inhibition of monocyte recruitment to the plaques	The plaque macrophage number decreased by 45% and the overall plaque load decreased by 28%	N/A	[190]
		PLK1	BI-2536	Inhibition of PLK1 → blocking of macrophage proliferation → reduce cell accumulation within the plaque	The proliferating macrophages (Ki67 +) were reduced by 60% and plaque stability was improved	N/A	[60]
		Anti-aging drugs	Dasatinib, Quercetin	Reduced inflammatory factor release in senescent macrophages	Elimination of senescent clonal hematopoietic cells → lighten SASP	N/A	[31]

## Data Availability

Publicly available datasets were analyzed in this study as described in Section.

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
