# Peer review of "Autophagy and Its Association with Macrophages in Clonal Hematopoiesis Leading to Atherosclerosis"

_ijms, 2025, doi:10.3390/ijms26073252_

Round 1
Reviewer 1 Report
Comments and Suggestions for Authors
Many Refs are missing
The last Section requires modifications
(The Authors must see my remarks)

Author Response
Reviewer 1:
Reviewer comment 1: “Comments and Suggestions for Authors: Many Refs are missing (The Authors must see my remarks)”
Author response to Reviewer comment 1: We thank the reviewer for this comment and the following specific suggestion. We have revised all your comments in the article one by one.
Reviewer comment 1-1: “State the type of the article....,eg. Review???” Comments and Suggestions from Reviewer’s note in PDF file “For R_reviewer 1_ijms-3474141-review”.
Author response to Reviewer comment 1-1: We thank the reviewer for this comment and the following specific suggestion. We have identified the type of article as “Review” (see Page1 Line 1).
Reviewer comment 1-2: “Refs???” (Original, Page 2 Lines 61-63) Comments and Suggestions from Reviewer’s note in PDF file “For R_reviewer 1_ijms-3474141-review ”.
Author response to Reviewer comment 1-2: We thank the reviewer for this comment and the following specific suggestion. We have added two references as “[10] Ajoolabady, A.; Pratico, D.; Lin, L.; Mantzoros, C. S.; Bahijri, S.; Tuomilehto, J.; Ren, J. (2024). Inflammation in atherosclerosis: pathophysiology and mechanisms. Cell Death Dis. 2024, 15, 817. doi: 10.1038/s41419-024-07166-8” and “[2] Pan, W.; Zhang, J.; Zhang, L.; Zhang, Y.; Song, Y.; Han, L.; Tan, M.; Yin, Y.; Yang, T.; Jiang, T.; et al. Comprehensive view of macrophage autophagy and its application in cardiovascular diseases. Cell Prolif. 2024, 57, e13525. doi: 10.1111/cpr.13525.” (in Red, Page 2 Line 67) also listed in Section References List (in Red, Page 20).
Reviewer comment 1-3: “Refs???” (Original, Page 2 Line 71) Comments and Suggestions from Reviewer’s note in PDF file “For R_reviewer 1_ijms-3474141-review”.
Author response to Reviewer comment 1-3: We thank the reviewer for this comment and the following specific suggestion. We have added one reference as “[5] Hou, P.; Fang, J.; Liu, Z., Shi, Y.; Agostini, M.; Bernassola, F.; Bove, P.; Candi, E.; Rovella, V.; Sica, G.; Sun, Q.; et al. Macrophage polarization and metabolism in atherosclerosis. Cell Death Dis. 2023, 14, 691. doi: 10.1038/s41419-023-06206-z.” (in Red, Page 2 Line 75) also listed in Section References List (in Red, Page 20).
Reviewer comment 1-4: “Such as???” (Original, Page 2 Lines 85-88) Comments and Suggestions from Reviewer’s note in PDF file “For R_reviewer 1_ijms-3474141-review”.
Author response to Reviewer comment 1-4: We thank the reviewer for this comment and the following specific suggestion. We have provided some examples for this sentence and added relevant references “Recent studies highlight pharmacological modulation of autophagy as a promising strategy to restore macrophage function and stabilize atherosclerotic plaques. For instance, metformin activates autophagy and suppresses NLRP3 inflammasome activity, reducing plaque progression [DX-14]. Chlorogenic acid enhances macrophage polarization and inflammation resolution by inducing autophagy, while rapamycin improves plaque stability by activating autophagy in macrophages [DX-15-DX-17]. Curcumin, a natural compound, also promotes autophagy in macrophages, alleviating atherosclerosis in animal models [DX-18]. These findings plus disused in this review suggest that targeting autophagy could offer potential therapeutic benefits in preventing or treating atherosclerosis by improving macrophage function and plaque stability” (in Blue Page 3, Lines 104-113). The following References were added in Section Reference List (in Red, Page 22):
[23] Tang, G.; Duan, F.; Li, W.; Wang, Y.; Zeng, C.; Hu, J.; Li, H.; Zhang, X.; Chen, Y.; Tan, H. Metformin inhibited Nod-like receptor protein 3 inflammasomes activation and suppressed diabetes-accelerated atherosclerosis in apoE-/- mice. Biomed Pharmacother. 2019, 119, 109410. doi: 10.1016/j.biopha.2019.109410.
[24] Gu, T.; Zhang, Z.; Liu, J.; Chen, L.; Tian, Y.; Xu, W.; Zeng, T.; Wu, W.; Lu, L. Chlorogenic Acid Alleviates LPS-Induced Inflammation and Oxidative Stress by Modulating CD36/AMPK/PGC-1α in RAW264.7 Macrophages. Int J Mol Sci. 2023, 24, 13516. doi: 10.3390/ijms241713516.
[25] Guo, Y.; Qin, J.; Zhao, Q.; Yang, J.; Wei, X.; Huang, Y.; Xie, M.; Zhang, C.; Li, Y. Plaque-Targeted Rapamycin Spherical Nucleic Acids for Synergistic Atherosclerosis Treatment. Adv Sci (Weinh). 2022, 9, e2105875. doi: 10.1002/advs.202105875.
[26] Grootaert, M. O. J.; Moulis, M.; Roth, L.; Martinet, W.; Vindis, C.; Bennett, M. R.; De Meyer, G. R. Y. Vascular smooth muscle cell death, autophagy and senescence in atherosclerosis. Cardiovasc Res. 2018, 114, 622-634. doi: 10.1093/cvr/cvy007.
[27] Li, X.; Zhu, R.; Jiang, H.; Yin, Q.; Gu, J.; Chen, J.; Ji, X.; Wu, X.; Fu, H.; Wang, H.; et al. Autophagy enhanced by curcumin ameliorates inflammation in atherogenesis via the TFEB-P300-BRD4 axis. Acta Pharm Sin B. 2022, 12, 2280-2299. doi: 10.1016/j.apsb.2021.12.014.
Reviewer comment 1-5: “Refs???” (Original, Page 3 Lines 85-88) Comments and Suggestions from Reviewer’s note in PDF file “For R_reviewer 1_ijms-3474141-review”.
Author response to Reviewer comment 1-5: We thank the reviewer for this comment and the following specific suggestion. Rational of this figure is summarized from all the references we discussed above, therefore we could not list all references here in Figure Legend.
Reviewer comment 1-6: “Such as???” (Original, Page 7 Lines 243-246) Comments and Suggestions from Reviewer’s note in PDF file “For R_reviewer 1_ijms-3474141-review”.
Author response to Reviewer comment 1-6: We thank the reviewer for this comment and the following specific suggestion. We have added one reference as “[29] Qiao, L.; Ma, J.; Zhang, Z.; Sui, W.; Zhai, C.; Xu, D.; Wang, Z.; Lu, H.;Zhang, M.; Zhang, C.; et al. Deficient Chaperone-Mediated Autophagy Promotes Inflammation and Atherosclerosis. Circ. Res. 2021, 129, 1141-1157. doi: 10.1161/CIRCRESAHA.121.318908” (in Red, Page9 Line 333) also listed in Section References List (in Red, Page 21).
Reviewer comment 1-7: “Refs????” (Original, Page 9 Lines 319-322) Comments and Suggestions from Reviewer’s note in PDF file “For R_reviewer 1_ijms-3474141-review”.
Author response to Reviewer comment 1-7: We thank the reviewer for this comment and the following specific suggestion. We have added two references as “[98] De Meyer, G. R. Y.; Zurek, M.; Puylaert, P.; Martinet, W. Programmed death of macrophages in atherosclerosis: mechanisms and therapeutic targets. Nat Rev Cardiol. 2024, 21, 312-325. doi: 10.1038/s41569-023-00957-0” and “[107] Liu, C.; Jiang, Z.; Pan, Z.; Yang, L. The Function, Regulation and Mechanism of Programmed Cell Death of Macrophages in Atherosclerosis. Front Cell Dev Biol. 2022, 9, 809516. doi: 10.3389/fcell.2021.809516” (in Red, Page 11 Line 409), and also listed in Section References List (in Red, Page 24).
Reviewer comment 1-8: “Refs????” (Original, Page 9 Lines 345-351) Comments and Suggestions from Reviewer’s note in PDF file “For R_reviewer 1_ijms-3474141-review”.
Author response to Reviewer comment 1-8: We thank the reviewer for this comment and the following specific suggestion. We have added citation [98]. “[121] Luo, X.; Weng, X.; Bao, X.; Bai, X.; Lv, Y.; Zhang, S.; Chen, Y.; Zhao, C.; Zeng, M.; Huang, J.; et al. A novel anti-atherosclerotic mechanism of quercetin: Competitive binding to KEAP1 via Arg483 to inhibit macrophage pyroptosis. Redox Biol. 2022, 57, 102511. doi: 10.1016/j.redox.2022.102511” (in Red, Page 10 Line 438) also listed in Section References List (in Red,Page 25).
Reviewer comment 1-9: “Refs????” (Original, Page 10 Lines 383-388) Comments and Suggestions from Reviewer’s note in PDF file “For R_reviewer 1_ijms-3474141-review”.
Author response to Reviewer comment 1-9: We thank the reviewer for this comment and the following specific suggestion. We have added two references as “[101] and “[126] Rocha, M.; Apostolova, N.; Diaz-Rua, R.; Muntane, J.; Victor, V. M. Mitochondria and T2D: Role of Autophagy, ER Stress, and Inflammasome. Trends Endocrinol Metab. 2020, 31, 725-741. doi: 10.1016/j.tem.2020.03.004” (in Red, Page 12 Lines 475-476), also listed in Section References List (in Red,Page 25).
Reviewer comment 1-10: “Refs????” (Original, Page 11 Lines 443-445) Comments and Suggestions from Reviewer’s note in PDF file “For R_reviewer 1_ijms-3474141-review”.
Author response to Reviewer comment 1-10: We thank the reviewer for this comment and the following specific suggestion. We have added two references as “[157] Kinzhebay, A.; Salybekov, A. A. The Role of Somatic Mutations in Ischemic Stroke: CHIP's Impact on Vascular Health. Neurology International. 2025, 17, 19. doi: 10.3390/neurolint17020019” and “[21] Cobo, I.; Tanaka, T.; Glass, C. K.; Yeang, C. Clonal hematopoiesis driven by DNMT3A and TET2 mutations: role in monocyte and macrophage biology and atherosclerotic cardiovascular disease. Curr Opin Hematol. 2022, 29, 1-7. doi: 10.1097/MOH.0000000000000688” (in Red, Page 14 Line 593) also listed in Section References List (in Red,Pages 21, 27).
Reviewer comment 1-11: “Remove the Year....” (Original, Page 11 Line 448) Comments and Suggestions from Reviewer’s note in PDF file “For R_reviewer 1_ijms-3474141-review”.
Author response to Reviewer comment 1-11: We thank the reviewer for this comment and the following specific suggestion. We have already removed the year “(2021)”. (Page 14 Line 597)
Reviewer comment 1-12: “Remove....” (Original, Page 11 Line 461) Comments and Suggestions from Reviewer’s note in PDF file “For R_reviewer 1_ijms-3474141-review”.
Author response to Reviewer comment 1-12: We thank the reviewer for this comment and the following specific suggestion. We have already removed the year “(2020)” (Page 15 Line 609). We also remove the year “(2017)” (Page 15 Line 620).
Reviewer comment 1-13: “Refs???” (Original, Page 13 Lines 520-530) Comments and Suggestions from Reviewer’s note in PDF file “For R_reviewer 1_ijms-3474141-review”.
Author response to Reviewer comment 1-13: We thank the reviewer for this comment and the following specific suggestion. We have added [131] (Page 17 Line 675).
Reviewer comment 1-14: “Remove....” (Original, Page 14 Line 558) Comments and Suggestions from Reviewer’s note in PDF file “For R_reviewer 1_ijms-3474141-review”.
Author response to Reviewer comment 1-14: We thank the reviewer for this comment and the following specific suggestion. We have already removed the year “(2021)”. (Page 17 Line 708)
Reviewer comment 1-15: “Refs???” (Original, Page 15 Lines 604-608) Comments and Suggestions from Reviewer’s note in PDF file “For R_reviewer 1_ijms-3474141-review”.
Author response to Reviewer comment 1-15: We thank the reviewer for this comment and the following specific suggestion. We have added two references as “[113]” (in Red, Page 15 Line 606) and “[186] Zhang, Q.; Zhao, L.; Yang, Y.; Li, S.; Liu, Y.; Chen, C. Mosaic loss of chromosome Y promotes leukemogenesis and clonal hematopoiesis. JCI Insight. 2022, 7, e153768. doi: 10.1172/jci” (in Red, Page 18 Line 757), and also listed in Section References List (in Red,Page 28).
Reviewer comment 1-16: “Refs???” in Section Conclusion and Perspective (Original, Page 15 Lines 604-608) Comments and Suggestions from Reviewer’s note in PDF file “For R_reviewer 1_ijms-3474141-review”.
Author response to Reviewer comment 1-16: We thank the reviewer for this comment and the following specific suggestion. Because we were asked removing all References in Section Conclusion and Perspective, we have added two into the paragraph before the section as “[187] Yamamoto, H.; Zhang, S.; Mizushima, N. Autophagy genes in biology and disease. Nat Rev Genet. 2023, 24, 382-400. doi: 10.1038/s41576-022-00562-w” and “[188] Levine, B.; Kroemer, G. Biological Functions of Autophagy Genes: A Disease Perspective. Cell. 2019 Jan 10;176(1-2):11-42. doi: 10.1016/j.cell.2018.09.048” (in Red, Page 15 Line 608), and also listed in Section References List (in Red,Page 28).
Reviewer comment 2: The last Section requires modifications: “Remove..... No Refs in that Section....Moreover, reduce that Section and state the main outcomes only...”
Author response to Reviewer comment 1: We appreciate the reviewer’s suggestion and have changed the last Section “Conclusion and Perspective” as suggested. (1) Almost all References were removed (we keep two references in the last paragraph because it is very necessary for our further discussion), (2) We shortened this section and only left the main outcomes this section. (3) And we eventually shorten this part to almost half size of the original, as “Atherosclerosis is a multifaceted, chronic inflammatory disease in which various proteolytic pathways, such as macroautophagy, the ubiquitin-proteasome system (UPS), and chaperone-mediated autophagy (CMA), play pivotal roles. Among these, macroautophagy has been well-characterized in atherosclerosis, while the contribution of CMA remains less understood. CMA is a selective protein degradation process that targets over 45% of cytoplasmic proteins containing KFERQ-like motifs, and it is crucial for maintaining cellular proteostasis and lysosomal function. Dysregulation of CMA can influence macroautophagy and exacerbate atherosclerosis progression. Notably, CMA facilitates the degradation of pro-atherogenic proteins, such as NLRP3 inflammasomes, making it a promising target for therapeutic intervention in atherosclerosis. Despite this potential, further research is required to fully understand how CMA affects NLRP3 inflammasome activity and its broader role in atherosclerotic pathogenesis.
CMA dysfunction, particularly in macrophages and their associated clonal hematopoiesis, results in lipid accumulation, oxidative stress, and inflammatory cytokine release—key contributors to atherosclerosis. While the link between CMA deficiency and disrupted lipid metabolism in macrophages is established, the precise mechanisms by which CMA regulates lipid accumulation and atherogenesis remain unclear. CMA also plays a vital role in lipid homeostasis by modulating lipid-regulating enzymes, thereby preventing foam cell formation and plaque instability. Emerging evidence suggests that both macroautophagy and CMA contribute to macrophage function and plaque stability, pointing to the potential for therapeutic strategies that target both pathways to mitigate atherosclerosis progression. These findings underscore the necessity of further research into the intricate regulation of lipid metabolism by CMA.
Another critical aspect of atherosclerosis is macrophage pyroptosis, a pro-inflammatory form of cell death that destabilizes plaques. Pyroptosis, mediated by proteins such as GSDMD, exacerbates atherosclerosis by impairing cholesterol transport, with GSDMD knockout models showing reduced lesion areas [163, 164]. Additionally, GSDME, another regulator of inflammatory cell death, can switch apoptosis to pyroptosis depending on caspase-3 activity. GSDME’s role in regulating cell death pathways presents a promising therapeutic target for atherosclerosis” (in Red, Pages 18-19, Lines 761-791), and “In conclusion, the intricate interplay between CMA, lipid metabolism, and pyroptosis represents an untapped opportunity for innovative therapeutic interventions. Targeting these pathways, either individually or synergistically, offers a powerful strategy to slow or even reverse the progression of atherosclerosis, with the potential to significantly impact the management of atherosclerotic cardiovascular disease” (in Red, Page 19, Lines 803-801).
(4) Because of revision suggestion from another Reviewer, we have added an extra paragraph as “Building on recent advances, we hypothesize that CMA, in conjunction with other forms of autophagy, programmed cell death, and clonal hematopoiesis, forms a crucial regulatory axis that governs all-source macrophage function in atherosclerosis. This network plays a pivotal role in regulating macrophage homeostasis, inflammation, and lipid metabolism, all of which are central to the development and progression of atherosclerosis. Disruption of this axis—whether through CMA dysfunction or clonal hematopoiesis—can lead to excessive inflammation, impaired lipid handling, and plaque destabilization. This hypothesis suggests that targeting the key components of this axis may offer a powerful approach for mitigating atherosclerosis and preventing its complications. Understanding the intricate relationships between autophagy, cell death, and clonal hematopoiesis will be essential in designing novel therapeutic interventions for atherosclerotic cardiovascular disease.” (in Blue, Page 19, Lines 791-802)
Reviewer 2 Report
Comments and Suggestions for Authors
The article “Autophagy and its Association with Macrophages in Clonal Hematopoiesis Leading to Atherosclerosis” submitted for review is part of the current trend of interest in autophagy and its involvement in the etiopathogenesis of a wide range of diseases. Corrections are required before the article is eligible for publication:
- A graphic abstract is recommended.
- The authors should explain what new contributions their article makes to the area of knowledge discussed.
- Please improve the quality of Figure 1. Use the Microsoft Word template or LaTeX template.
- Please characterize the macrophages including the origin of vascular macrophages.
- Please discuss the potential and risk of pharmacological manipulation of autophagic pathways.
Author Response
Journal: International Journal of Molecular Sciences
Manuscript Status: Pending major revisions
Manuscript ID: ijms-3474141
Section: Molecular Pathology, Diagnostics, and Therapeutics
Special Issue: Cellular and Molecular Mechanisms of Myocardial Diseases
Title: Autophagy and its Association with Macrophages in Clonal Hematopoiesis leading to Atherosclerosis
Authorship: Shuanhu Li, Xhin Zhou, ... , Xuehong Xu, Shou-Ping Gong, Huiling Cao
Dear Reviewers and Editor-in-Chief,
According to the Comments and Suggestions from reviewers, we have responded to them all. All changes are listed here item-by-item and are highlighted in Color Letter in our revised manuscript.
We appreciate your effort and time!
Xuehong Xu, MS/PhD.
Reviewer 2:
Comments and Suggestions for Authors: The article “Autophagy and its Association with Macrophages in Clonal Hematopoiesis Leading to Atherosclerosis” submitted for review is part of the current trend of interest in autophagy and its involvement in the etiopathogenesis of a wide range of diseases. Corrections are required before the article is eligible for publication:
Author response to reviewer main comment: We thank the reviewer for this comment on the importance of this topic described and the following specific suggestion. We will make item-by-item corrections by following the Comments and Suggestions in revised manuscript.
Reviewer comment 1: A graphic abstract is recommended.
Author response to Reviewer comment 1: We thank the reviewer for this specific comment on our manuscript and have added a graphic abstract.
Reviewer comment 2: The authors should explain what new contributions their article makes to the area of knowledge discussed.
Author response to Reviewer comment 2: We thank the reviewer for this comment on our manuscript. A hypothesis we proposed in ABSTRACT as “Building upon current advances, we propose a hypothesis in which autophagy, programmed cell death, and clonal hematopoiesis form a critical intrinsic axis that modulates the fundamental functions of macrophages, playing a complex role in the development of atherosclerosis” (in Red, Page 1, Lines 37-40), which are the main logic linking throughout entire manuscript. Therefore, this intellectual concept would be mentioned wherever including the insertion in the last paragraph in Section INTRODUCTION as “Building upon recent advances in the field, we hypothesize that autophagy, cellular programmed cell death, and clonal hematopoiesis form a critical intrinsic axis, modulating the basic functions of macrophages, which in turn plays a multifaceted and complex role in the development and progression of atherosclerosis. Specifically, this axis is hypothesized to regulate macrophage homeostasis, inflammatory responses, and lipid metabolism, all of which are key drivers in atherosclerotic plaque formation and plaque instability. Moreover, autophagic processes in macrophages are essential not only for cellular maintenance and inflammation resolution but also for preventing exaggerated immune responses and cellular senescence. When this axis is dysregulated, as seen in the context of autophagy defects or somatic mutations associated with clonal hematopoiesis, macrophages are prone to heightened inflammatory states and abnormal lipid accumulation, leading to accelerated atherosclerosis progression. These insights suggest that a more integrated understanding of the molecular underpinnings linking autophagy and clonal hematopoiesis could reveal new therapeutic targets for managing atherosclerotic cardiovascular disease” (in Red, Pages 2-3, Lines 89-104).
We also logically corporate the concept into Section “2. Chaperone-Mediated Autophagy (CMA) initiates in Inflammation and atherosclerosis” as “Expanding on recent findings, in conjunction with other forms of autophagy, programmed cell death, and clonal hematopoiesis, establishes an intricate regulatory network that governs macrophage function in atherosclerosis. This network plays a critical role in macrophage homeostasis, inflammatory regulation, and lipid metabolism, all of which are essential for the development and progression of atherosclerosis. Dysregulation of this axis—whether through impaired CMA, autophagic dysfunction, or mutations related to clonal hematopoiesis—can amplify inflammation and lipid accumulation, accelerating disease progression. These insights highlight the potential for CMA as a therapeutic target for modulating inflammation and plaque stability, offering novel opportunities” (in Red, Page 4, Lines 135-145).
We also logically corporate the concept into Section “Conclusion and Perspective” as “Building on recent advances, we hypothesize that CMA, in conjunction with other forms of autophagy, programmed cell death, and clonal hematopoiesis, forms a crucial regulatory axis that governs macrophage function in atherosclerosis. This network plays a pivotal role in regulating macrophage homeostasis, inflammation, and lipid metabolism, all of which are central to the development and progression of atherosclerosis. Disruption of this axis—whether through CMA dysfunction or clonal hematopoiesis—can lead to excessive inflammation, impaired lipid handling, and plaque destabilization. This hypothesis suggests that targeting the key components of this axis may offer a powerful approach for mitigating atherosclerosis and preventing its complications. Understanding the intricate relationships between autophagy, cell death, and clonal hematopoiesis will be essential in designing novel therapeutic interventions for atherosclerotic cardiovascular disease” (in Blue, Page 19, Lines 791-802.
Reviewer comment 3: Please improve the quality of Figure 1. Use the Microsoft Word template or LaTeX template.
Author response to Reviewer comment 3: We thank the reviewer for this comment on our manuscript and we have improved the quality of Figure 1 (Page 3 Line 107, Figure 1).Also we also uploaded online.
Reviewer comment 4: Please characterize the macrophages including the origin of vascular macrophages.
Author response to Reviewer comment 4: We thank the reviewer for this comment and specific suggestion. We have added a sub-section “3.1. Macrophages and their fundamental functions” after the Section “3. Deficient Chaperone-Mediated Autophagy Promotes Lipid Accumulation in Macro-phage” as “3.1. Macrophages and their fundemental functions
Macrophages are highly versatile immune cells that perform a wide range of functions, critical to both homeostasis and disease pathology. They are able to adopt distinct phenotypic and functional states depending on the signals present in their local microenvironment. As central players in the immune response, macrophages maintain tissue homeostasis, regulate inflammation, and contribute to tissue repair by engulfing pathogens, cellular debris, and apoptotic cells, as well as secreting a diverse array of cytokines to modulate immune reactions [50].
Morphologically, macrophages are large, irregularly shaped cells with a highly adaptable, amoeboid form. Their cytoplasm is rich in organelles, including lysosomes, mitochondria, and rough endoplasmic reticulum, which are essential for their phagocytic activity and various other cellular processes. These cells are identified by surface markers such as CD14, CD40, CD11b, CD64, F4/80, EMR1, lysozyme M, MAC-1/MAC-3, and CD68, which serve to define their unique role in immune surveillance and response [51,52].
Macrophages exhibit remarkable plasticity, being classified into two primary polarization states: M1 and M2. M1 macrophages are typically induced by pro-inflammatory stimuli such as lipopolysaccharides (LPS) and interferon-gamma (IFN-γ). These cells are characterized by the production of pro-inflammatory cytokines, including tumor necrosis factor-alpha (TNF-α) and interleukin-1 beta (IL-1β), as well as reactive oxygen species (ROS), all of which contribute to inflammation and tissue damage [53]. On the other hand, M2 macrophages are driven by anti-inflammatory signals like interleukin-4 (IL-4) and IL-13, and they are primarily involved in tissue repair, wound healing, and the resolution of inflammation. M2 macrophages release anti-inflammatory cytokines such as IL-10 and facilitate the clearance of apoptotic cells [53].
Furthermore, macrophages exhibit functional diversity based on their anatomical location and activation status. For example, alveolar macrophages in the lungs play a crucial role in clearing inhaled pathogens and particulate matter, thereby contributing to pulmonary defense [54,55]. Kupffer cells in the liver are responsible for detoxifying metabolic byproducts and removing toxins from the bloodstream [56]. Microglia in the central nervous system monitor brain homeostasis, clear dead cells, and respond to injury. Inflammatory macrophages, recruited to sites of infection or injury, are classified into M1 and M2 subtypes based on their functional profiles and cytokine production [57,58].
Vascular macrophages represent a specialized subset of macrophages that reside in or are closely associated with blood vessels, where they play key roles in maintaining vascular integrity and homeostasis. These cells originate from multiple sources, including yolk sac progenitors, bone marrow-derived monocytes, and tissue-resident progenitors [59]. The developmental origins of vascular macrophages are complex, with embryonic yolk sac-derived CX3CR1+ endothelial microparticles (EMPs) and fetal liver monocytes contributing to the establishment of the tissue-resident macrophage population in the arterial wall during early development [60]. Vascular macrophages can be further classified into distinct subtypes, including M1, M2, Mox, M4, and Mhem macrophages, each with unique gene expression profiles and functional roles. For example, Mox macrophages are involved in heme detoxification and mitigating oxidative stress, while M4 macrophages produce chemokines and proteases that recruit additional immune cells and degrade extracellular matrix components [61,62].
In response to vascular injury, inflammation, or pathological conditions such as atherosclerosis or vascular calcification, circulating monocytes infiltrate the vessel wall and differentiate into macrophages. These macrophages play critical roles in vascular pathology, including foam cell formation in atherosclerotic plaques [63], promoting vascular calcification through the secretion of osteogenic factors [64], and facilitating tissue repair through the clearance of apoptotic cells and resolution of inflammation[65]. The plasticity of macrophages allows them to adapt to a variety of microenvironments, making them essential regulators of vascular health and disease” (in Red Pages 6-7 Lines 233-285).
Corresponding to this additional Sub-section, we have also cited 16 references in Section References List (in Red, Page 21).
Reviewer comment 5: Please discuss the potential and risk of pharmacological manipulation of autophagic pathways.
Author response to Reviewer comment 5: We thank the reviewer for this comment and specific suggestion. We have added a sub-section “4.4. Potential and Risk of Pharmacological Manipulation of Autophagic Pathways” after the Sub-section “4.3. Mechanistic Insights and Therapeutic Targets of Autophagy-Mediated Macrophage Pyroptosis” as “4.4. Potential and Risk of Pharmacological Manipulation of Autophagic Pathways
As autophagy is an essential cellular degradation process through which damaged organelles, proteins, and other intracellular debris are sequestered in autophagosomes and degraded by lysosomal enzymes, thereby maintaining cellular homeostasis, pharmacological modulation of the autophagy pathway has garnered attention for its potential therapeutic applications across various diseases in recent years. However, this approach also carries inherent risks that require careful consideration.
In the context of atherosclerosis, autophagy plays a crucial role in both the progression and resolution of the disease. Pharmacologically activating autophagy in macrophages, such as through the use of rapamycin, has been shown to promote the polarization of an anti-inflammatory phenotype, reduce vascular inflammation, and improve vascular remodeling [132]. Furthermore, autophagy plays a key role in lipid metabolism, influencing the formation and resolution of atherosclerotic plaques. For example, autophagic activation can reduce plaque progression by facilitating the removal of lipid droplets within macrophages. In patients with non-alcoholic fatty liver disease (NAFLD), autophagy activation has been demonstrated to alleviate hepatic lipid accumulation. Resveratrol, a small molecule drug, can activate the AMPK pathway and inhibit mTORC1 activity, thereby promoting autophagic flux and regulating autophagy-related gene expression via the PI3K/AKT signaling pathway [133].
Beyond lipid metabolism, autophagy is integral to cellular homeostasis, particularly in neurons. For instance, autophagic activation reduces the accumulation of α-synuclein in the brains of Parkinson's disease models [134]. Additionally, the autophagy-inducing drug trehalose has demonstrated protective effects in animal models of neurodegenerative diseases, highlighting its potential for clinical application [135].
Autophagy also has a dual role in cancer therapy. On one hand, autophagy can promote cancer cell survival and resistance to chemotherapy. On the other hand, autophagy modulation, such as through rapamycin, may enhance the efficacy of cancer therapies. However, excessive autophagy activation can lead to cellular dysfunction, including disrupted energy metabolism and the induction of apoptotic pathways [136]. Dysregulated autophagy may also contribute to lipotoxicity, thereby increasing oxidative stress within cells. In certain contexts, excessive autophagy can exacerbate disease progression, such as in tumors where autophagy supports growth and drug resistance.
While the pharmacological manipulation of autophagy pathways holds considerable therapeutic promise, it is not without its risks. Overactivation of autophagy can impair cellular function, leading to metabolic disturbances and heightened oxidative stress. Conversely, inhibition of autophagy may result in an exacerbated inflammatory response, potentially aggravating disease progression. For instance, chronic administration of rapamycin, although effective in inducing autophagy, has been associated with immunosuppression and metabolic disorders, underscoring the need for caution in its long-term use [137,138].
In summary, while pharmacological regulation of the autophagy pathway holds significant promise for the treatment of various diseases, it is essential to evaluate the risks and fine-tune therapeutic strategies to ensure efficacy and minimize adverse effects. Future research should focus on elucidating the precise mechanisms governing autophagy and developing more selective and safer pharmacological agents that can harness the full therapeutic potential of autophagy without undesirable side effects” (in Red, Pages 12-13, Lines 503-548).
Corresponding to this additional Sub-section, we have also added 16 references in Section References List (in Red, Page 20).
Reviewer 3 Report
Comments and Suggestions for Authors
For years, it has been proposed that macrophage autophagy plays a protective role in advanced atherosclerosis. Li et al. discuss the association between autophagy and macrophages in clonal hematopoiesis, which is believed to contribute to atherosclerosis. This article is comprehensive and well-conceived; however, some of the cited references are over a decade old. To be considered for publication in a high-impact journal, the authors must make significant revisions, beginning with an updated reference list.
Additionally, the objectives and aims of the study are not clearly articulated. The current conclusion appears to be merely a condensed discussion, resulting in an article that feels fragmented and lacking a specific aim. A concise, clear take-home message is needed in the conclusion.
A significant shortcoming of this review is the review itself. Without a defined methodology and rigorous scrutiny typical of a systematic review, this paper reads more like an anthology of existing works on the topic.Authors should define the search terms, specify databases used, and clearly annotate the inclusion and exclusion criteria.
Author Response
Journal: International Journal of Molecular Sciences
Manuscript Status: Pending major revisions
Manuscript ID: ijms-3474141
Title: Autophagy and its Association with Macrophages in Clonal Hematopoiesis leading to Atherosclerosis
Authorship: Shuanhu Li, Xhin Zhou, ... , Xuehong Xu, Shou-Ping Gong, Huiling Cao
Dear Reviewers and Editor-in-Chief,
According to the Comments and Suggestions from reviewers, we have responded to them all. All changes are listed here item-by-item and are highlighted in Color Letter in our revised manuscript.
We appreciate your effort and time!
Xuehong Xu, MS/PhD.
Reviewer 3:
Comments and Suggestions for Authors
For years, it has been proposed that macrophage autophagy plays a protective role in advanced atherosclerosis. Li et al. discuss the association between autophagy and macrophages in clonal hematopoiesis, which is believed to contribute to atherosclerosis. This article is comprehensive and well-conceived; however, some of the cited references are over a decade old. To be considered for publication in a high-impact journal, the authors must make significant revisions, beginning with an updated reference list.
Author response to Reviewer comment 1: We thank the reviewer for this comment on our manuscript. Based on suggestions, we updated a large number of references in Section References List (in Blue and Red, Page 20). We have revised our manuscript according to all your comments and suggestions in our following responses.
Author response to Reviewer comment 2: “Additionally, the objectives and aims of the study are not clearly articulated. The current conclusion appears to be merely a condensed discussion, resulting in an article that feels fragmented and lacking a specific aim. A concise, clear take-home message is needed in the conclusion.”
Author response to Reviewer comment 2: We thank the reviewer for this comment on our manuscript and we have revised the conclusion section. (Page 15-16, Lines 609-667)
- We have added the concepts of this rearview work in different locations as “Building upon current advances, we propose a hypothesis in which autophagy, programmed cell death, and clonal hematopoiesis form a critical intrinsic axis that modulates the fundamental functions of macrophages, playing a complex role in the development of atherosclerosis” (in Red, Page 1, Lines 37-40), which are the main logic linking throughout entire manuscript. Therefore, this intellectual concept would be mentioned wherever including the insertion in the last paragraph in Section INTRODUCTION as “Building upon recent advances in the field, we hypothesize that autophagy, cellular programmed cell death, and clonal hematopoiesis form a critical intrinsic axis, modulating the basic functions of macrophages, which in turn plays a multifaceted and complex role in the development and progression of atherosclerosis. Specifically, this axis is hypothesized to regulate macrophage homeostasis, inflammatory responses, and lipid metabolism, all of which are key drivers in atherosclerotic plaque formation and plaque instability. Moreover, autophagic processes in macrophages are essential not only for cellular maintenance and inflammation resolution but also for preventing exaggerated immune responses and cellular senescence. When this axis is dysregulated, as seen in the context of autophagy defects or somatic mutations associated with clonal hematopoiesis, macrophages are prone to heightened inflammatory states and abnormal lipid accumulation, leading to accelerated atherosclerosis progression. These insights suggest that a more integrated understanding of the molecular underpinnings linking autophagy and clonal hematopoiesis could reveal new therapeutic targets for managing atherosclerotic cardiovascular disease” (in Red, Pages 2-3, Lines 89-104).
- We also logically corporate the concept into Section “ Chaperone-Mediated Autophagy (CMA) initiates in Inflammation and atherosclerosis” as “Expanding on recent findings, in conjunction with other forms of autophagy, programmed cell death, and clonal hematopoiesis, establishes an intricate regulatory network that governs macrophage function in atherosclerosis. This network plays a critical role in macrophage homeostasis, inflammatory regulation, and lipid metabolism, all of which are essential for the development and progression of atherosclerosis. Dysregulation of this axis—whether through impaired CMA, autophagic dysfunction, or mutations related to clonal hematopoiesis—can amplify inflammation and lipid accumulation, accelerating disease progression. These insights highlight the potential for CMA as a therapeutic target for modulating inflammation and plaque stability, offering novel opportunities” (in Red, Page 4, Lines 135-145).
- We also logically corporate the concept into Section “Conclusion and Perspective” as “Building on recent advances, we hypothesize that CMA, in conjunction with other forms of autophagy, programmed cell death, and clonal hematopoiesis, forms a crucial regulatory axis that governs macrophage function in atherosclerosis. This network plays a pivotal role in regulating macrophage homeostasis, inflammation, and lipid metabolism, all of which are central to the development and progression of atherosclerosis. Disruption of this axis—whether through CMA dysfunction or clonal hematopoiesis—can lead to excessive inflammation, impaired lipid handling, and plaque destabilization. This hypothesis suggests that targeting the key components of this axis may offer a powerful approach for mitigating atherosclerosis and preventing its complications. Understanding the intricate relationships between autophagy, cell death, and clonal hematopoiesis will be essential in designing novel therapeutic interventions for atherosclerotic cardiovascular disease” (in Blue, Page 19, Lines 791-802).
Author response to Reviewer comment 3: “A significant shortcoming of this review is the review itself. Without a defined methodology and rigorous scrutiny typical of a systematic review, this paper reads more like an anthology of existing works on the topic. Authors should define the search terms, specify databases used, and clearly annotate the inclusion and exclusion criteria.”
Author response to Reviewer comment 3: We thank the reviewer for this comment and specific suggestion. According to the comments and suggestions, we have made much more modification/re-writing in this revised manuscript as the below:
(1) We have added two extra Sub-sections. The first section is the below to make the macrophage concept and it fundamental function more clear, as “3.1. Macrophages and their fundamental functions” after the Section “3. Deficient Chaperone-Mediated Autophagy Promotes Lipid Accumulation in Macro-phage” as “as “3.1. Macrophages and their fundemental functions
Macrophages are highly versatile immune cells that perform a wide range of functions, critical to both homeostasis and disease pathology. They are able to adopt distinct phenotypic and functional states depending on the signals present in their local microenvironment. As central players in the immune response, macrophages maintain tissue homeostasis, regulate inflammation, and contribute to tissue repair by engulfing pathogens, cellular debris, and apoptotic cells, as well as secreting a diverse array of cytokines to modulate immune reactions [50].
Morphologically, macrophages are large, irregularly shaped cells with a highly adaptable, amoeboid form. Their cytoplasm is rich in organelles, including lysosomes, mitochondria, and rough endoplasmic reticulum, which are essential for their phagocytic activity and various other cellular processes. These cells are identified by surface markers such as CD14, CD40, CD11b, CD64, F4/80, EMR1, lysozyme M, MAC-1/MAC-3, and CD68, which serve to define their unique role in immune surveillance and response [51,52].
Macrophages exhibit remarkable plasticity, being classified into two primary polarization states: M1 and M2. M1 macrophages are typically induced by pro-inflammatory stimuli such as lipopolysaccharides (LPS) and interferon-gamma (IFN-γ). These cells are characterized by the production of pro-inflammatory cytokines, including tumor necrosis factor-alpha (TNF-α) and interleukin-1 beta (IL-1β), as well as reactive oxygen species (ROS), all of which contribute to inflammation and tissue damage [53]. On the other hand, M2 macrophages are driven by anti-inflammatory signals like interleukin-4 (IL-4) and IL-13, and they are primarily involved in tissue repair, wound healing, and the resolution of inflammation. M2 macrophages release anti-inflammatory cytokines such as IL-10 and facilitate the clearance of apoptotic cells [53].
Furthermore, macrophages exhibit functional diversity based on their anatomical location and activation status. For example, alveolar macrophages in the lungs play a crucial role in clearing inhaled pathogens and particulate matter, thereby contributing to pulmonary defense [54,55]. Kupffer cells in the liver are responsible for detoxifying metabolic byproducts and removing toxins from the bloodstream [56]. Microglia in the central nervous system monitor brain homeostasis, clear dead cells, and respond to injury. Inflammatory macrophages, recruited to sites of infection or injury, are classified into M1 and M2 subtypes based on their functional profiles and cytokine production [57,58].
Vascular macrophages represent a specialized subset of macrophages that reside in or are closely associated with blood vessels, where they play key roles in maintaining vascular integrity and homeostasis. These cells originate from multiple sources, including yolk sac progenitors, bone marrow-derived monocytes, and tissue-resident progenitors [59]. The developmental origins of vascular macrophages are complex, with embryonic yolk sac-derived CX3CR1+ endothelial microparticles (EMPs) and fetal liver monocytes contributing to the establishment of the tissue-resident macrophage population in the arterial wall during early development [60]. Vascular macrophages can be further classified into distinct subtypes, including M1, M2, Mox, M4, and Mhem macrophages, each with unique gene expression profiles and functional roles. For example, Mox macrophages are involved in heme detoxification and mitigating oxidative stress, while M4 macrophages produce chemokines and proteases that recruit additional immune cells and degrade extracellular matrix components [61,62].
In response to vascular injury, inflammation, or pathological conditions such as atherosclerosis or vascular calcification, circulating monocytes infiltrate the vessel wall and differentiate into macrophages. These macrophages play critical roles in vascular pathology, including foam cell formation in atherosclerotic plaques [63], promoting vascular calcification through the secretion of osteogenic factors [64], and facilitating tissue repair through the clearance of apoptotic cells and resolution of inflammation[65]. The plasticity of macrophages allows them to adapt to a variety of microenvironments, making them essential regulators of vascular health and disease” (in Red Pages 6-7 Lines 233-285).
Corresponding to this additional Sub-section, we have also added 16 references in Section References List (in Red, Page 20).
(2) The second Sub-section as “4.4. Potential and Risk of Pharmacological Manipulation of Autophagic Pathways” after the Sub-section “4.3. Mechanistic Insights and Therapeutic Targets of Autophagy-Mediated Macrophage Pyroptosis”. In this sub-section, we have provided our analysis on the possibility and risk of the autophagy as as “4.4. Potential and Risk of Pharmacological Manipulation of Autophagic Pathways
As autophagy is an essential cellular degradation process through which damaged organelles, proteins, and other intracellular debris are sequestered in autophagosomes and degraded by lysosomal enzymes, thereby maintaining cellular homeostasis, pharmacological modulation of the autophagy pathway has garnered attention for its potential therapeutic applications across various diseases in recent years. However, this approach also carries inherent risks that require careful consideration.
In the context of atherosclerosis, autophagy plays a crucial role in both the progression and resolution of the disease. Pharmacologically activating autophagy in macrophages, such as through the use of rapamycin, has been shown to promote the polarization of an anti-inflammatory phenotype, reduce vascular inflammation, and improve vascular remodeling [132]. Furthermore, autophagy plays a key role in lipid metabolism, influencing the formation and resolution of atherosclerotic plaques. For example, autophagic activation can reduce plaque progression by facilitating the removal of lipid droplets within macrophages. In patients with non-alcoholic fatty liver disease (NAFLD), autophagy activation has been demonstrated to alleviate hepatic lipid accumulation. Resveratrol, a small molecule drug, can activate the AMPK pathway and inhibit mTORC1 activity, thereby promoting autophagic flux and regulating autophagy-related gene expression via the PI3K/AKT signaling pathway [133].
Beyond lipid metabolism, autophagy is integral to cellular homeostasis, particularly in neurons. For instance, autophagic activation reduces the accumulation of α-synuclein in the brains of Parkinson's disease models [134]. Additionally, the autophagy-inducing drug trehalose has demonstrated protective effects in animal models of neurodegenerative diseases, highlighting its potential for clinical application [135].
Autophagy also has a dual role in cancer therapy. On one hand, autophagy can promote cancer cell survival and resistance to chemotherapy. On the other hand, autophagy modulation, such as through rapamycin, may enhance the efficacy of cancer therapies. However, excessive autophagy activation can lead to cellular dysfunction, including disrupted energy metabolism and the induction of apoptotic pathways [136]. Dysregulated autophagy may also contribute to lipotoxicity, thereby increasing oxidative stress within cells. In certain contexts, excessive autophagy can exacerbate disease progression, such as in tumors where autophagy supports growth and drug resistance.
While the pharmacological manipulation of autophagy pathways holds considerable therapeutic promise, it is not without its risks. Overactivation of autophagy can impair cellular function, leading to metabolic disturbances and heightened oxidative stress. Conversely, inhibition of autophagy may result in an exacerbated inflammatory response, potentially aggravating disease progression. For instance, chronic administration of rapamycin, although effective in inducing autophagy, has been associated with immunosuppression and metabolic disorders, underscoring the need for caution in its long-term use [137,138].
In summary, while pharmacological regulation of the autophagy pathway holds significant promise for the treatment of various diseases, it is essential to evaluate the risks and fine-tune therapeutic strategies to ensure efficacy and minimize adverse effects. Future research should focus on elucidating the precise mechanisms governing autophagy and developing more selective and safer pharmacological agents that can harness the full therapeutic potential of autophagy without undesirable side effects” (in Red, Pages 12-13, Lines 503-548).
Corresponding to this additional Sub-section, we have also added 16 references in Section References List (in Red, Page 21).
(3) We have also re-written the Section of Conclusion and Perpective with more clear logic and it to almost half size of the original, as “Atherosclerosis is a multifaceted, chronic inflammatory disease in which various proteolytic pathways, such as macroautophagy, the ubiquitin-proteasome system (UPS), and chaperone-mediated autophagy (CMA), play pivotal roles. Among these, macroautophagy has been well-characterized in atherosclerosis, while the contribution of CMA remains less understood. CMA is a selective protein degradation process that targets over 45% of cytoplasmic proteins containing KFERQ-like motifs, and it is crucial for maintaining cellular proteostasis and lysosomal function. Dysregulation of CMA can influence macroautophagy and exacerbate atherosclerosis progression. Notably, CMA facilitates the degradation of pro-atherogenic proteins, such as NLRP3 inflammasomes, making it a promising target for therapeutic intervention in atherosclerosis. Despite this potential, further research is required to fully understand how CMA affects NLRP3 inflammasome activity and its broader role in atherosclerotic pathogenesis.
CMA dysfunction, particularly in macrophages and their associated clonal hematopoiesis, results in lipid accumulation, oxidative stress, and inflammatory cytokine release—key contributors to atherosclerosis. While the link between CMA deficiency and disrupted lipid metabolism in macrophages is established, the precise mechanisms by which CMA regulates lipid accumulation and atherogenesis remain unclear. CMA also plays a vital role in lipid homeostasis by modulating lipid-regulating enzymes, thereby preventing foam cell formation and plaque instability. Emerging evidence suggests that both macroautophagy and CMA contribute to macrophage function and plaque stability, pointing to the potential for therapeutic strategies that target both pathways to mitigate atherosclerosis progression. These findings underscore the necessity of further research into the intricate regulation of lipid metabolism by CMA.
Another critical aspect of atherosclerosis is macrophage pyroptosis, a pro-inflammatory form of cell death that destabilizes plaques. Pyroptosis, mediated by proteins such as GSDMD, exacerbates atherosclerosis by impairing cholesterol transport, with GSDMD knockout models showing reduced lesion areas [163, 164]. Additionally, GSDME, another regulator of inflammatory cell death, can switch apoptosis to pyroptosis depending on caspase-3 activity. GSDME’s role in regulating cell death pathways presents a promising therapeutic target for atherosclerosis” (in Red, Pages 18-19, Lines 761-791), and “In conclusion, the intricate interplay between CMA, lipid metabolism, and pyroptosis represents an untapped opportunity for innovative therapeutic interventions. Targeting these pathways, either individually or synergistically, offers a powerful strategy to slow or even reverse the progression of atherosclerosis, with the potential to significantly impact the management of atherosclerotic cardiovascular disease” (in Red, Page 19, Lines803-807).
Because of revision suggestion from another Reviewer, we have added an extra paragraphas “Building on recent advances, we hypothesize that CMA, in conjunction with other forms of autophagy, programmed cell death, and clonal hematopoiesis, forms a crucial regulatory axis that governs all-source macrophage function in atherosclerosis. This network plays a pivotal role in regulating macrophage homeostasis, inflammation, and lipid metabolism, all of which are central to the development and progression of atherosclerosis. Disruption of this axis—whether through CMA dysfunction or clonal hematopoiesis—can lead to excessive inflammation, impaired lipid handling, and plaque destabilization. This hypothesis suggests that targeting the key components of this axis may offer a powerful approach for mitigating atherosclerosis and preventing its complications. Understanding the intricate relationships between autophagy, cell death, and clonal hematopoiesis will be essential in designing novel therapeutic interventions for atherosclerotic cardiovascular disease.” (in Blue, Page 19, Lines 791-802).
(4) To accomplish the linking axis of Macrophage-Clonal Hematopoiesis-Autophagy, we have added extra paragraph in Section 5 as “In adult mammals, circulating macrophages, derived from the monocyte-macrophage lineage, are generated from four primary sources: (1) the yolk sac at embryonic day 7.0 (E7.0), (2) the embryonic liver at E9.5, (3) the dorsal aorta, and (4) the bone marrow. These sources give rise to pre-macrophages/macrophages, monocytes, perivascular stromal cells (PDGFRA+), and hematopoietic stem cells (HSCs) through distinct waves of hematopoiesis [131,133]. Upon migration, circulating macrophages can localize to various tissues, including the heart, where atrioventricular (AV) node-resident macrophages directly influence cardiac function, as characterized by electrocardiogram (EKG) abnormalities [130,132,135]. Genetic defects in macrophage function can lead to severe cardiac dysfunction, particularly by disrupting the integrity of the AV node and its associated structures, such as the plane of insulation (POI) adjacent to the central fibrous body (CFB) [132,135]. It is plausible that circulating macrophages with impaired autophagic function localize to the aortic wall, where they play a critical role in the pathogenesis of atherosclerosis” (Page 14 Lines 552-564).
We have added some extra terms as “AV node/AVN: atrioventricular node” (Page 14 Line 548), “CFB: central fibrous body” (Page 14 Line 552), “EKG: electrocardiogram” (Page 13 Line 549), “HSCs: hematopoietic stem cells” (Page 13 Line 546), “POI: plane of insulation” (Page 14 Line 552). Those were also added in Abbreviation List (Page 19 Lines 809-824).
Reviewer 4 Report
Comments and Suggestions for Authors
The authors have submitted a review article of illustrating a current knowledge regarding impact of a chaperon-mediated autophagy (CMA) on pathology of atherosclerosis which has been thought to be a result from the dysfunctions of lipid metabolism, endothelial permeability and macroautophagy. The authors searched a range of eligible literature, from well-known classical, and latest research regarding an association of immunological dysfunction with possible pathogenesis resulting in atherosclerosis, which might be primarily attributed to the pathogenesis of the disease. The authors discussed the beneficial availability of CMA which might ameliorate the states of the disease situation, resulting in reliable perspectives. This issue is of interest because of the fact that humans are widely affected with chronic inflammatory disorders such as atherosclerosis, and impact of their review is strong. My overall concern with the review describing the current available data regarding beneficial availability of CMA against the molecular mechanisms towards chronic inflammatory disorder is that information provided may offer something substantial that helps advance our understanding of effective management which draws novel treatment procedures available in clinic. The reference list may be useful for readers who are interested in this issue.
To strengthen authors’ perspectives, the authors are strongly recommended to add the expanded discussion which includes how to modulate CMA activity clinically. Also, please consider a “toxicology” sub-section regarding known activated CMA effect on humans, for instance. The opposite, toxicological effects of expected outcomes, if known, may influence largely the authors’ perspective for the effective treatment for chronic inflammatory disorders such as atherosclerosis.
Author Response
Journal: International Journal of Molecular Sciences
Manuscript Status: Pending major revisions
Manuscript ID: ijms-3474141
Title: Autophagy and its Association with Macrophages in Clonal Hematopoiesis leading to Atherosclerosis
Authorship: Shuanhu Li, Xhin Zhou, ... , Xuehong Xu, Shou-Ping Gong, Huiling Cao
Dear Reviewers and Editor-in-Chief,
According to the Comments and Suggestions from reviewers, we have responded to them all. All changes are listed here item-by-item and are highlighted in Color Letter in our revised manuscript.
We appreciate your effort and time!
Xuehong Xu, MS/PhD.
Reviewer 4:
Comments and Suggestions for Authors: “The authors have submitted a review article of illustrating a current knowledge regarding impact of a chaperon-mediated autophagy (CMA) on pathology of atherosclerosis which has been thought to be a result from the dysfunctions of lipid metabolism, endothelial permeability and macroautophagy. The authors searched a range of eligible literature, from well-known classical, and latest research regarding an association of immunological dysfunction with possible pathogenesis resulting in atherosclerosis, which might be primarily attributed to the pathogenesis of the disease. The authors discussed the beneficial availability of CMA which might ameliorate the states of the disease situation, resulting in reliable perspectives. This issue is of interest because of the fact that humans are widely affected with chronic inflammatory disorders such as atherosclerosis, and impact of their review is strong. My overall concern with the review describing the current available data regarding beneficial availability of CMA against the molecular mechanisms towards chronic inflammatory disorder is that information provided may offer something substantial that helps advance our understanding of effective management which draws novel treatment procedures available in clinic. The reference list may be useful for readers who are interested in this issue.”
Author Response to main Reviewer Comment: We thank the reviewers for their comments on the importance and relevance of this topic described in our manuscript and for their specific suggestions.
Reviewer comment 1: “To strengthen authors’ perspectives, the authors are strongly recommended to add the expanded discussion which includes how to modulate CMA activity clinically.”
Author response to Reviewer comment 1: We thank the reviewers for their comments on this specific suggestion. We have added an extra discussion and our opinion as “Expanding on recent findings, in conjunction with other forms of autophagy, programmed cell death, and clonal hematopoiesis, establishes an intricate regulatory network that governs macrophage function in atherosclerosis. This network plays a critical role in macrophage homeostasis, inflammatory regulation, and lipid metabolism, all of which are essential for the development and progression of atherosclerosis. Dysregulation of this axis—whether through impaired CMA, autophagic dysfunction, or mutations related to clonal hematopoiesis—can amplify inflammation and lipid accumulation, accelerating disease progression. These insights highlight the potential for CMA as a therapeutic target for modulating inflammation and plaque stability, offering novel opportunities” (in Red, Page 4, Lines 135-145).
Reviewer comment 2: “Also, please consider a “toxicology” sub-section regarding known activated CMA effect on humans, for instance. The opposite, toxicological effects of expected outcomes, if known, may influence largely the authors’ perspective for the effective treatment for chronic inflammatory disorders such as atherosclerosis.”
Author response to Reviewer comment 2: We thank the reviewers for their comments on this specific suggestion. We have added an extra sub-section located on the basic fundamental characteristics of macrophages, which can help researchers on understanding possible CMA activated “toxicology” on human as “3.1. Macrophages and their fundemental functions
Macrophages are highly versatile immune cells that perform a wide range of functions, critical to both homeostasis and disease pathology. They are able to adopt distinct phenotypic and functional states depending on the signals present in their local microenvironment. As central players in the immune response, macrophages maintain tissue homeostasis, regulate inflammation, and contribute to tissue repair by engulfing pathogens, cellular debris, and apoptotic cells, as well as secreting a diverse array of cytokines to modulate immune reactions [50].
Morphologically, macrophages are large, irregularly shaped cells with a highly adaptable, amoeboid form. Their cytoplasm is rich in organelles, including lysosomes, mitochondria, and rough endoplasmic reticulum, which are essential for their phagocytic activity and various other cellular processes. These cells are identified by surface markers such as CD14, CD40, CD11b, CD64, F4/80, EMR1, lysozyme M, MAC-1/MAC-3, and CD68, which serve to define their unique role in immune surveillance and response [51,52].
Macrophages exhibit remarkable plasticity, being classified into two primary polarization states: M1 and M2. M1 macrophages are typically induced by pro-inflammatory stimuli such as lipopolysaccharides (LPS) and interferon-gamma (IFN-γ). These cells are characterized by the production of pro-inflammatory cytokines, including tumor necrosis factor-alpha (TNF-α) and interleukin-1 beta (IL-1β), as well as reactive oxygen species (ROS), all of which contribute to inflammation and tissue damage [53]. On the other hand, M2 macrophages are driven by anti-inflammatory signals like interleukin-4 (IL-4) and IL-13, and they are primarily involved in tissue repair, wound healing, and the resolution of inflammation. M2 macrophages release anti-inflammatory cytokines such as IL-10 and facilitate the clearance of apoptotic cells [53].
Furthermore, macrophages exhibit functional diversity based on their anatomical location and activation status. For example, alveolar macrophages in the lungs play a crucial role in clearing inhaled pathogens and particulate matter, thereby contributing to pulmonary defense [54,55]. Kupffer cells in the liver are responsible for detoxifying metabolic byproducts and removing toxins from the bloodstream [56]. Microglia in the central nervous system monitor brain homeostasis, clear dead cells, and respond to injury. Inflammatory macrophages, recruited to sites of infection or injury, are classified into M1 and M2 subtypes based on their functional profiles and cytokine production [57,58].
Vascular macrophages represent a specialized subset of macrophages that reside in or are closely associated with blood vessels, where they play key roles in maintaining vascular integrity and homeostasis. These cells originate from multiple sources, including yolk sac progenitors, bone marrow-derived monocytes, and tissue-resident progenitors [59]. The developmental origins of vascular macrophages are complex, with embryonic yolk sac-derived CX3CR1+ endothelial microparticles (EMPs) and fetal liver monocytes contributing to the establishment of the tissue-resident macrophage population in the arterial wall during early development [60]. Vascular macrophages can be further classified into distinct subtypes, including M1, M2, Mox, M4, and Mhem macrophages, each with unique gene expression profiles and functional roles. For example, Mox macrophages are involved in heme detoxification and mitigating oxidative stress, while M4 macrophages produce chemokines and proteases that recruit additional immune cells and degrade extracellular matrix components [61,62].
In response to vascular injury, inflammation, or pathological conditions such as atherosclerosis or vascular calcification, circulating monocytes infiltrate the vessel wall and differentiate into macrophages. These macrophages play critical roles in vascular pathology, including foam cell formation in atherosclerotic plaques [63], promoting vascular calcification through the secretion of osteogenic factors [64], and facilitating tissue repair through the clearance of apoptotic cells and resolution of inflammation[65]. The plasticity of macrophages allows them to adapt to a variety of microenvironments, making them essential regulators of vascular health and disease” (in Red Pages 6-7 Lines 233-285).
Corresponding to this additional Sub-section, we have also added 16 references in Section References List (in Red, Page 20).
Reviewer 5 Report
Comments and Suggestions for Authors
The paper entitled: “Autophagy and its Association with Macrophages in Clonal Hematopoiesis leading to Atherosclerosis” by Shuanhu Li et al. deals with a review of macrophage autophagy in atherosclerosis processes.
The authors summarize various mechanistic elements, such as the importance of autophagic regulation in macrophages, focusing on its role in inflammation, plaque formation, and the contributions of clonal hematopoiesis.
Clonal hematopoiesis, another process, and somatic mutations in genes like TET2, JAK2, and DNMT3A that drive immune cell expansion, can enhance inflammatory responses in atherosclerotic plaques.
In fact, the mechanistic framework in Atherosclerosis pathology is very complex.
Their description is apparently narrative leading to a core hypothesis: that regulating gasdermin E mediated pyroptosis may shift cell death modes across various cell types based on its expression profile, although its regulatory mechanisms warrant further investigation.
A diagram depicting the molecular structure of the specific macrophage mediated targeted mechanism could be useful to help the reader and at a glance the fruition of the paper.
Since this is the core hypothesis of this work we suggest it is presented beforehand and the whole development then follows.
Maybe, shifting paragraphs in order to start from the broad and go into the details of macrophages' role.
Therapeutic strategies directed to these specific mechanisms in atherosclerosis are just hinted at. We suggest they present and evaluate how current treatments in the clinics can play in these mechanisms.
A table summarizing current treatments and a table summarizing clinical trials about novel proposed molecules could be useful. At least a table summarizing preclinical results about proposed molecules should be added.
Author Response
Journal: International Journal of Molecular Sciences
Manuscript Status: Pending major revisions
Manuscript ID: ijms-3474141
Title: Autophagy and its Association with Macrophages in Clonal Hematopoiesis leading to Atherosclerosis
Authorship: Shuanhu Li, Xhin Zhou, ... , Xuehong Xu, Shou-Ping Gong, Huiling Cao
Dear Reviewers and Editor-in-Chief,
According to the Comments and Suggestions from reviewers, we have responded to them all. All changes are listed here item-by-item and are highlighted in Color Letter in our revised manuscript.
We appreciate your effort and time!
Xuehong Xu, MS/PhD.
Reviewer 5:
Comments and Suggestions for Authors
The paper entitled: “Autophagy and its Association with Macrophages in Clonal Hematopoiesis leading to Atherosclerosis” by Shuanhu Li et al. deals with a review of macrophage autophagy in atherosclerosis processes. The authors summarize various mechanistic elements, such as the importance of autophagic regulation in macrophages, focusing on its role in inflammation, plaque formation, and the contributions of clonal hematopoiesis. Clonal hematopoiesis, another process, and somatic mutations in genes like TET2, JAK2, and DNMT3A that drive immune cell expansion, can enhance inflammatory responses in atherosclerotic plaques.
Author Response to main Reviewer Comment: We thank the reviewers for their comments on the importance and relevance of this topic described in our manuscript and for their specific suggestions.
Reviewer comment 1: “In fact, the mechanistic framework in Atherosclerosis pathology is very complex.”
Author response to Reviewer comment 1: We strongly agree with this opinion of the reviewer on the topics. The mechanistic framework in Atherosclerosis pathology is even much more complicated than that we discussed through entire manuscript. We along with other researchers will keep working and focusing on it for a long period of time.
Reviewer comment 2: “Their description is apparently narrative leading to a core hypothesis: that regulating gasdermin E mediated pyroptosis may shift cell death modes across various cell types based on its expression profile, although its regulatory mechanisms warrant further investigation. ”
Author response to Reviewer comment 2: We thank the reviewers for the comments. The core hypothesis- that regulating gasdermin E mediated pyroptosis may shift cell death modes across various cell types based on its expression profile is very positive as a breaking point that unveil critical targeting for developing therapeutics treatment or diagnostics at least.
Reviewer comment 3: “A diagram depicting the molecular structure of the specific macrophage mediated targeted mechanism could be useful to help the reader and at a glance the fruition of the paper. Since this is the core hypothesis of this work we suggest it is presented beforehand and the whole development then follows. Maybe, shifting paragraphs in order to start from the broad and go into the details of macrophages' role.”
Author response to Reviewer comment 3: We thank the reviewers for their comments. We have added an extra sub-section located on the basic fundamental characteristics of macrophages. We believe that can help researchers on this issue as as “3.1. Macrophages and their fundemental functions
Macrophages are highly versatile immune cells that perform a wide range of functions, critical to both homeostasis and disease pathology. They are able to adopt distinct phenotypic and functional states depending on the signals present in their local microenvironment. As central players in the immune response, macrophages maintain tissue homeostasis, regulate inflammation, and contribute to tissue repair by engulfing pathogens, cellular debris, and apoptotic cells, as well as secreting a diverse array of cytokines to modulate immune reactions [50].
Morphologically, macrophages are large, irregularly shaped cells with a highly adaptable, amoeboid form. Their cytoplasm is rich in organelles, including lysosomes, mitochondria, and rough endoplasmic reticulum, which are essential for their phagocytic activity and various other cellular processes. These cells are identified by surface markers such as CD14, CD40, CD11b, CD64, F4/80, EMR1, lysozyme M, MAC-1/MAC-3, and CD68, which serve to define their unique role in immune surveillance and response [51,52].
Macrophages exhibit remarkable plasticity, being classified into two primary polarization states: M1 and M2. M1 macrophages are typically induced by pro-inflammatory stimuli such as lipopolysaccharides (LPS) and interferon-gamma (IFN-γ). These cells are characterized by the production of pro-inflammatory cytokines, including tumor necrosis factor-alpha (TNF-α) and interleukin-1 beta (IL-1β), as well as reactive oxygen species (ROS), all of which contribute to inflammation and tissue damage [53]. On the other hand, M2 macrophages are driven by anti-inflammatory signals like interleukin-4 (IL-4) and IL-13, and they are primarily involved in tissue repair, wound healing, and the resolution of inflammation. M2 macrophages release anti-inflammatory cytokines such as IL-10 and facilitate the clearance of apoptotic cells [53].
Furthermore, macrophages exhibit functional diversity based on their anatomical location and activation status. For example, alveolar macrophages in the lungs play a crucial role in clearing inhaled pathogens and particulate matter, thereby contributing to pulmonary defense [54,55]. Kupffer cells in the liver are responsible for detoxifying metabolic byproducts and removing toxins from the bloodstream [56]. Microglia in the central nervous system monitor brain homeostasis, clear dead cells, and respond to injury. Inflammatory macrophages, recruited to sites of infection or injury, are classified into M1 and M2 subtypes based on their functional profiles and cytokine production [57,58].
Vascular macrophages represent a specialized subset of macrophages that reside in or are closely associated with blood vessels, where they play key roles in maintaining vascular integrity and homeostasis. These cells originate from multiple sources, including yolk sac progenitors, bone marrow-derived monocytes, and tissue-resident progenitors [59]. The developmental origins of vascular macrophages are complex, with embryonic yolk sac-derived CX3CR1+ endothelial microparticles (EMPs) and fetal liver monocytes contributing to the establishment of the tissue-resident macrophage population in the arterial wall during early development [60]. Vascular macrophages can be further classified into distinct subtypes, including M1, M2, Mox, M4, and Mhem macrophages, each with unique gene expression profiles and functional roles. For example, Mox macrophages are involved in heme detoxification and mitigating oxidative stress, while M4 macrophages produce chemokines and proteases that recruit additional immune cells and degrade extracellular matrix components [61,62].
In response to vascular injury, inflammation, or pathological conditions such as atherosclerosis or vascular calcification, circulating monocytes infiltrate the vessel wall and differentiate into macrophages. These macrophages play critical roles in vascular pathology, including foam cell formation in atherosclerotic plaques [63], promoting vascular calcification through the secretion of osteogenic factors [64], and facilitating tissue repair through the clearance of apoptotic cells and resolution of inflammation[65]. The plasticity of macrophages allows them to adapt to a variety of microenvironments, making them essential regulators of vascular health and disease” (in Red Pages 6-7 Lines 233-285).
Corresponding to this additional Sub-section, we have also added 16 references in Section References List (in Red, Page 20).
Reviewer comment 4: “Therapeutic strategies directed to these specific mechanisms in atherosclerosis are just hinted at. We suggest they present and evaluate how current treatments in the clinics can play in these mechanisms.”
Author response to Reviewer comment 4: We thank the reviewers for this specific comments and suggestions. We have added an extra sub-section as “4.4. Potential and Risk of Pharmacological Manipulation of Autophagic Pathways”, in which we discussed some current treatments and their risk based on the molecular mechanism of selective and non-selective autophagy as as “4.4. Potential and Risk of Pharmacological Manipulation of Autophagic Pathways
As autophagy is an essential cellular degradation process through which damaged organelles, proteins, and other intracellular debris are sequestered in autophagosomes and degraded by lysosomal enzymes, thereby maintaining cellular homeostasis, pharmacological modulation of the autophagy pathway has garnered attention for its potential therapeutic applications across various diseases in recent years. However, this approach also carries inherent risks that require careful consideration.
In the context of atherosclerosis, autophagy plays a crucial role in both the progression and resolution of the disease. Pharmacologically activating autophagy in macrophages, such as through the use of rapamycin, has been shown to promote the polarization of an anti-inflammatory phenotype, reduce vascular inflammation, and improve vascular remodeling [132]. Furthermore, autophagy plays a key role in lipid metabolism, influencing the formation and resolution of atherosclerotic plaques. For example, autophagic activation can reduce plaque progression by facilitating the removal of lipid droplets within macrophages. In patients with non-alcoholic fatty liver disease (NAFLD), autophagy activation has been demonstrated to alleviate hepatic lipid accumulation. Resveratrol, a small molecule drug, can activate the AMPK pathway and inhibit mTORC1 activity, thereby promoting autophagic flux and regulating autophagy-related gene expression via the PI3K/AKT signaling pathway [133].
Beyond lipid metabolism, autophagy is integral to cellular homeostasis, particularly in neurons. For instance, autophagic activation reduces the accumulation of α-synuclein in the brains of Parkinson's disease models [134]. Additionally, the autophagy-inducing drug trehalose has demonstrated protective effects in animal models of neurodegenerative diseases, highlighting its potential for clinical application [135].
Autophagy also has a dual role in cancer therapy. On one hand, autophagy can promote cancer cell survival and resistance to chemotherapy. On the other hand, autophagy modulation, such as through rapamycin, may enhance the efficacy of cancer therapies. However, excessive autophagy activation can lead to cellular dysfunction, including disrupted energy metabolism and the induction of apoptotic pathways [136]. Dysregulated autophagy may also contribute to lipotoxicity, thereby increasing oxidative stress within cells. In certain contexts, excessive autophagy can exacerbate disease progression, such as in tumors where autophagy supports growth and drug resistance.
While the pharmacological manipulation of autophagy pathways holds considerable therapeutic promise, it is not without its risks. Overactivation of autophagy can impair cellular function, leading to metabolic disturbances and heightened oxidative stress. Conversely, inhibition of autophagy may result in an exacerbated inflammatory response, potentially aggravating disease progression. For instance, chronic administration of rapamycin, although effective in inducing autophagy, has been associated with immunosuppression and metabolic disorders, underscoring the need for caution in its long-term use [137,138].
In summary, while pharmacological regulation of the autophagy pathway holds significant promise for the treatment of various diseases, it is essential to evaluate the risks and fine-tune therapeutic strategies to ensure efficacy and minimize adverse effects. Future research should focus on elucidating the precise mechanisms governing autophagy and developing more selective and safer pharmacological agents that can harness the full therapeutic potential of autophagy without undesirable side effects” (in Red, Pages 12-13, Lines 503-548).
Corresponding to this additional Sub-section, we have also added 8 references in Section References List (in Red, Page 20).
Reviewer comment 5: “A table summarizing current treatments and a table summarizing clinical trials about novel proposed molecules could be useful. At least a table summarizing preclinical results about proposed molecules should be added.”
Author response to Reviewer comment 5: We thank the reviewers for this specific comments and suggestions. We have added a table (Table-2) summarizing current treatments and clinical trials based on the molecular mechanism of selective and non-selective autophagy as “Table 2 Current Treatments and Preclinical/Clinical Data for Atherosclerosis” (Pages 13, Lines 549-550).
Round 2
Reviewer 2 Report
Comments and Suggestions for Authors
The authors have improved the manuscript and I recommend it for further processing.
Reviewer 3 Report
Comments and Suggestions for Authors
I find myself uncertain about the contributions of this study, similar to my previous submission. The authors should highlight the potential strengths of this research that extend beyond existing knowledge. The objective of the paper is unclear, and the conclusions are vague and inconsistent.
Reviewer 4 Report
Comments and Suggestions for Authors
The authors have done a very good job responding to reviewer comments and concerns in their revision. I believe the manuscript is significantly improved as a result. Now I recommend that this revised version of the manuscript can be accepted for publication in IJMS.
Reviewer 5 Report
Comments and Suggestions for Authors
In the revised version of the manuscript the authors have addressed extensively and convincingly our previous criticisms. We think the paper can now be considerd ready for publication.